# Risk-Averse Total-Reward Reinforcement Learning

**Xihong Su**
Department of Computer Science
University of New Hampshire
Durham, NH 03824
xihong.su@unh.edu

**Jia Lin Hau**
Department of Computer Science
University of New Hampshire
Durham, NH 03824
jialin.hau@unh.edu

**Gersi Doko**
Department of Computer Science
University of New Hampshire
Durham, NH 03824
gersi.doko@unh.edu

**Kishan Panaganti**
Department of Computing & Mathematical Sciences
California Institute of Technology
(now at Tencent AI Lab, Seattle, WA)
kpb.research@gmail.com

**Marek Petrik**
Department of Computer Science
University of New Hampshire
Durham, NH 03824
marek.petrik@unh.edu

## Abstract

Risk-averse total-reward Markov Decision Processes (MDPs) offer a promising framework for modeling and solving undiscounted infinite-horizon objectives. Existing model-based algorithms for risk measures like the entropic risk measure (ERM) and entropic value-at-risk (EVaR) are effective in small problems, but require full access to transition probabilities. We propose a Q-learning algorithm to compute the optimal stationary policy for total-reward ERM and EVaR objectives with strong convergence and performance guarantees. The algorithm and its optimality are made possible by ERM's dynamic consistency and elicitability. Our numerical results on tabular domains demonstrate quick and reliable convergence of the proposed Q-learning algorithm to the optimal risk-averse value function.

## 1 Introduction

Risk-averse reinforcement learning (RL) is an essential framework for practical and high-stakes applications, such as self-driving, robotic surgery, healthcare, and finance [6, 12, 16, 17, 19, 22, 25, 26, 29, 30, 34, 41, 42, 45, 47]. A risk-averse policy prefers actions with more certainty even if it means a lower expected return [30, 38]. The goal of risk-averse RL is to compute risk-averse policies. To this end, risk-averse RL specifies the objective using monetary risk measures, such as value-at-risk (VaR), conditional value-at-risk (CVaR), entropic risk measure (ERM), or entropic value-at-risk (EVaR). These risk measures penalize the variability of returns interpretably and yield policies with stronger guarantees on the probability of catastrophic losses [14].

A major challenge in deriving practical RL algorithms is that the model of the environment is often unknown. This challenge is even more salient in risk-averse RL because computing the risk of random return involves evaluating the full distribution of the return rather than just its expectation [17, 40]. Traditional definitions of risk measures assume a known discounted [6, 18, 19, 22, 25, 26, 29] or transient [44] MDP model and use learning to approximate a global value or policy function [32].

Recent works have presented model-free methods to find optimal CVaR policies [23, 27] or optimal VaR policies [17], where the optimal policy may be Markovian or history-dependent. Another recent work derives a risk-sensitive Q-learning algorithm, which applies a nonlinear utility function to the temporal difference (TD) residual [38].

In RL domains formulated as Markov Decision Processes (MDPs), future rewards are handled differently based on whether the model is discounted or undiscounted. Most RL formulations assume discounted infinite-horizon objectives in which future rewards are assigned a lower value than present rewards. Discounting in financial domains is typically justified by interest rates and inflation, which make earlier rewards inherently more valuable [20, 36]. However, many RL tasks, such as robotics or games, have no natural justification for discounting. Instead, the domains have absorbing terminal states, which may represent desirable or undesirable outcomes [4, 15, 28].

Total reward criterion (TRC) generalizes stochastic shortest and longest path problems and has gained more attention [1, 9, 12, 13, 21, 31, 43, 44]. TRC has absorbing goal states and does not discount future rewards. A common assumption is that the MDP is transient and guarantees that any policy eventually terminates with positive probability. While this assumption is sufficient to guarantee finite risk-neutral expected returns, it is insufficient to guarantee finite returns with risk-averse objectives. For example, given an ERM objective, the risk level factor must also be sufficiently small to ensure a finite return [35, 44].

This paper derives risk-averse TRC model-free Q-learning algorithms for ERM-TRC and EVaR-TRC objectives. There are three main challenges to deriving these risk-averse TRC Q-learning algorithms. First, generalizing the standard Q-learning algorithm to risk-averse objectives requires computing the full return distribution rather than just its expectation. Second, the risk-averse TRC Bellman operator may not be a contraction. The absence of a contraction property precludes the direct application of model-free methods in an undiscounted MDP. Third, instead of the contraction property, we need to rely on other properties of the Bellman operator and an additional bounded condition to prove the convergence of the risk-averse TRC Q-learning algorithms.

As our main contribution, we derive a rigorous proof of the ERM-TRC Q-learning algorithm and the EVaR-TRC Q-learning algorithm converging to the optimal risk-averse value functions. We also show that the proposed Q-learning algorithms compute the optimal stationary policies for the ERM-TRC and EVaR-TRC objectives, and the optimal state-action value function can be computed by stochastic gradient descent along the derivative of the exponential loss function.

The rest of the paper is organized as follows. We define our research setting and preliminary concepts in Section 2. In Section 3, we leverage the elicitability of ERM, define a new ERM Bellman operator, and propose Q-learning algorithms for the ERM and EVaR objectives. In Section 4, we give a rigorous convergence proof of the proposed ERM-TRC Q-learning and EVaR-TRC Q-learning algorithms. Finally, the numerical results presented in Section 5 illustrate the effectiveness of our algorithms.

## 2 Preliminaries for Risk-aversion in Markov Decision Processes

We first introduce our notations and overview relevant properties for monetary risk measures. We then formalize the MDP framework with the risk-averse objective and summarize the standard Q-learning algorithm.

**Notation**  We denote by $\mathbb{R}$ and $\mathbb{N}$ the sets of real and natural (including $0$) numbers, and $\bar{\mathbb{R}} := \mathbb{R} \cup \{-\infty, \infty\}$ denotes the extended real line. We use $\mathbb{R}_+$ and $\mathbb{R}_{++}$ to denote non-negative and positive real numbers, respectively. We use a tilde to indicate a random variable, such as $\tilde{x} \colon \Omega \to \mathbb{R}$ for the sample space $\Omega$. The set of all real-valued random variables is denoted as $\mathbb{X} := \mathbb{R}^{\Omega}$. Sets are denoted with calligraphic letters.

**Monetary risk measures**  Monetary risk measures generalize the expectation operator to account for the uncertainty of the random variable. In this work, we focus on two risk measures. The first one is the *entropic risk measure* (ERM) defined for any risk level $\beta > 0$ and $\tilde{x} \in \mathbb{X}$ as [14]

$$\mathrm{ERM}_\beta\left[\tilde{x}\right] := -\beta^{-1} \cdot \log \mathbb{E} \exp\left(-\beta \cdot \tilde{x}\right), \tag{1}$$

and can be extended to $\beta \in [0, \infty]$ as $\mathrm{ERM}_0[\tilde{x}] = \lim_{\beta \to 0^+} \mathrm{ERM}_\beta[\tilde{x}] = \mathbb{E}[\tilde{x}]$ and $\mathrm{ERM}_\infty[\tilde{x}] = \lim_{\beta \to \infty} \mathrm{ERM}_\beta[\tilde{x}] = \mathrm{ess\,inf}[\tilde{x}]$. ERM is popular because of its simplicity and its favorable proper-

ties in multi-stage optimization formulations [17–19, 44]. In particular, dynamic decision-making with ERM allows for the existence of dynamic programming equations and Markov or stationary optimal policies. In addition, in this work, we leverage the fact that ERM is *elicitable*, which means that it can be estimated by solving a linear regression problem [7, 11].

The second risk measure we consider is *entropic value-at-risk* (EVaR), which is defined for a given risk level $\alpha \in (0, 1)$ and $\tilde{x} \in \mathbb{X}$ as

$$\text{EVaR}_\alpha\left[\tilde{x}\right] := \sup_{\beta > 0} -\beta^{-1} \log\left(\alpha^{-1} \mathbb{E} \exp\left(-\beta\tilde{x}\right)\right) = \sup_{\beta > 0} \text{ERM}_\beta\left[\tilde{x}\right] + \beta^{-1} \log \alpha, \quad (2)$$

and is extended to $\text{EVaR}_0\left[\tilde{x}\right] = \text{ess inf}[\tilde{x}]$ and $\text{EVaR}_1\left[\tilde{x}\right] = \mathbb{E}[\tilde{x}]$ [2]. It is important to note that the supremum in (2) may not be attained even when $\tilde{x}$ is a finite discrete random variable [3]. EVaR addresses several important shortcomings of ERM [17–19, 44]. In particular, EVaR is coherent and closely approximates popular and interpretable quantile-based risk measures, like VaR and CVaR [2, 19].

**Risk-Averse Markov Decision Processes**    We formulate the decision process as a *Markov Decision Process* (MDP) $(\mathcal{S}, \mathcal{A}, p, r, \mu)$ [36]. The set $\mathcal{S} = \{1, 2, \ldots, S, e\}$ is the finite set of states and $e$ represents a *sink* state. The set $\mathcal{A} = \{1, 2, \ldots, A\}$ is the finite set of actions. The transition function $p\colon \mathcal{S} \times \mathcal{A} \to \Delta_\mathcal{S}$ represents the probability $p(s, a, s')$ of transitioning to $s' \in \mathcal{S}$ after taking $a \in \mathcal{A}$ in $s \in \mathcal{S}$. The function $r\colon \mathcal{S} \times \mathcal{A} \times \mathcal{S} \to \mathbb{R}$ represents the reward $r(s, a, s') \in \mathbb{R}$ associated with transitioning from $s \in \mathcal{S}$ and $a \in \mathcal{A}$ to $s' \in \mathcal{S}$. The vector $\mu \in \Delta_\mathcal{S}$ is the initial state distribution.

As the objective, we focus on computing *stationary deterministic policies* $\Pi := \mathcal{A}^\mathcal{S}$ that maximize the *total reward criterion* (TRC):

$$\max_{\pi \in \Pi} \lim_{t \to \infty} \text{Risk}^{\pi,\mu}\left[\sum_{k=0}^{t-1} r(\tilde{s}_k, \tilde{a}_k, \tilde{s}_{k+1})\right], \quad (3)$$

where Risk represents either ERM or EVaR risk measures. The limit in (3) exists for stationary policies $\pi$ but may be infinite [44]. The superscript $\pi$ indicates the policy that guides the actions' probability, and the $\mu$ indicates the distribution over the initial states.

Recent work has shown that an optimal policy in (3) exists, is stationary, and has bounded return for a sufficiently small $\beta$ as long as the following assumptions hold [44]. The sink state $e$ satisfies that $p(e, a, e) = 1$ and $r(e, a, e) = 0$ for each $a \in \mathcal{A}$, and $\mu_e = 0$. It is crucial to assume that the MDP is *transient* for any $\pi \in \Pi$:

$$\sum_{t=0}^\infty \mathbb{P}^{\pi,s}\left[\tilde{s}_t = s'\right] < \infty, \qquad \forall s, s' \in \mathcal{S} \setminus \{e\}. \quad (4)$$

Intuitively, this assumption states that any policy eventually reaches the sink state and effectively terminates. We adopt these assumptions in the remainder of the paper.

With a risk-neutral objective, transient MDPs guarantee that the return is bounded for each policy. However, that is no longer the case with the ERM objective. In particular, for some $\beta$ it is possible that the return is unbounded. The return is bounded for a sufficiently small $\beta$ and there exists an optimal stationary policy [44]. In contrast, the return of EVaR is always bounded, and an optimal stationary policy always exists regardless of the risk level $\alpha$ [39].

**Standard Q-learning**    To help with the exposition of our new algorithms, we now informally summarize the Q-learning algorithm for a risk-neutral setting [4]. Q-learning is an essential component of most model-free reinforcement learning algorithms, including DQN and many actor-critic methods [10, 24, 46]. Its simplicity and scalability make it especially appealing.

Q-learning iteratively refines an estimate of the optimal state-action value function $\tilde{q}_i\colon \mathcal{S} \times \mathcal{A} \to \mathbb{R}, i \in \mathbb{N}$ such that it satisfies

$$\tilde{q}_{i+1}(\tilde{s}_i, \tilde{a}_i) = \tilde{q}_i(\tilde{s}_i, \tilde{a}_i) - \tilde{\eta}_i \tilde{z}_i, \quad \tilde{z}_i = r(\tilde{s}_i, \tilde{a}_i, \tilde{s}'_i) + \max_{a' \in \mathcal{A}} \tilde{q}_i(\tilde{s}'_i, a') - \tilde{q}_i(\tilde{s}_i, \tilde{a}_i). \quad (5)$$

Here, $\tilde{z}_i$ is also known as the TD residual. The algorithm assumes a stream of samples $(\tilde{s}_i, \tilde{a}_i, \tilde{s}'_i)_{i=1,\ldots}$ sampled from the transition probabilities and appropriately chosen step sizes $\tilde{\eta}_i$ to converge to the

optimal state-action value function. Note that $\tilde{q}_i$ is random because it is a function of a random variable. We restate lemma C.13 from [17] as the following lemma because it shows the connection between Q-learning and gradient descent.

**Lemma 2.1** (Lemma C.13 in [17]). *Suppose that $f : \mathbb{R} \to \mathbb{R}$ is a differentiable $\mu-$strongly convex function with an $L-$Lipschitz continuous gradient. Consider $x_i \in \mathbb{R}$ and a gradient update for any step size $\xi \in (0, 1/L]$ :*

$$x_{i+1} := x_i - \xi \cdot f'(x_i).$$

*Then $\exists l \in [1/L, 1/\mu]$ such that $\xi/l \in (0, 1]$ and*

$$x_{i+1} = (1 - \xi/l) \cdot x_i + \xi/l \cdot x^\star,$$

*where $x^\star = \arg\min_{x \in \mathbb{R}} f(x)$ is unique from the strong convexity of $f$.*

The standard Q-learning algorithm can be seen as a stochastic gradient descent on the quadratic loss function $f$ [5, 17, 38]. We leverage this property to build our algorithms.

# 3 Q-learning Algorithms: ERM and EVaR

In this section, we derive new Q-learning algorithms for the ERM and EVaR objectives. First, we propose the Q-learning algorithm for ERM in Section 3.1, which requires us to introduce a new ERM Bellman operator based on the elicitability of the ERM risk measure. Then, we use this algorithm in Section 3.2 to propose an EVaR Q-learning algorithm. The proofs for this section are deferred to Appendix A.

## 3.1 ERM Q-learning Algorithm

The algorithm we propose in this section computes the state-action value function for multiple values of the risk level $\beta \in \mathcal{B}$ for some given non-empty *finite* set $\mathcal{B} \subseteq \mathbb{R}_{++}$. The set $\mathcal{B}$ may be a singleton or may include multiple values, which is necessary to optimize EVaR in the next section.

Before describing the Q-learning algorithm, we define the ERM risk-averse state-action value function and describe the Bellman operator that can be used to compute it. We define the ERM risk-averse state-action value function $q_\beta : \mathcal{S} \times \mathcal{A} \times \mathcal{B} \to \bar{\mathbb{R}}$ for each $\beta \in \mathcal{B}$ as

$$q^\pi(s, a, \beta) := \lim_{t \to \infty} \text{ERM}_\beta^{\pi, \langle s, a \rangle} \left[ \sum_{k=0}^{t-1} r(\tilde{s}_k, \tilde{a}_k, \tilde{s}_{k+1}) \right], \quad q^\star(s, a, \beta) := \max_{\pi \in \Pi} q^\pi(s, a, \beta). \quad (6)$$

for each $s \in \mathcal{S}, a \in \mathcal{A}, \beta \in \mathcal{B}$. The limit in this equation exists by [43, lemma D.5].

The superscript $\langle s, a \rangle$ in (6) indicates the initial state and action are $\tilde{s}_0 = s, \tilde{a}_0 = a$, and the policy $\pi$ determines the actions henceforth. We use $\mathcal{Q} := \mathbb{R}^{\mathcal{S} \times \mathcal{A}}$ to denote the set of possible state-action value functions.

To facilitate the computation of the value functions, we define the following ERM Bellman operator $B_\beta : \mathbb{R}^{\mathcal{S} \times \mathcal{A}} \to \mathbb{R}^{\mathcal{S} \times \mathcal{A}}$ as

$$(B_\beta q)(s, a) := \text{ERM}_\beta^{a, s} \left[ r(s, a, \tilde{s}_1) + \max_{a' \in \mathcal{A}} q(\tilde{s}_1, a', \beta) \right], \quad \forall s \in \mathcal{S}, a \in \mathcal{A}, \beta \in \mathcal{B}, q \in \mathcal{Q}. \quad (7)$$

Note that the Bellman operator $B_\beta$ applies to the value functions $q(\cdot, \cdot, \beta)$ for a fixed value of $\beta$. We abbreviate $B_\beta$ to $B$ when the risk level $\beta$ is clear from the context.

The following result, which follows from Theorem A.1, shows that the ERM state-action value function can be computed as the fixed point of the Bellman operator.

**Theorem 3.1.** *Assume some $\beta \in \mathcal{B}$ and suppose that $q_\beta^\star(s, a, \beta) > -\infty, \forall s \in \mathcal{S}, a \in \mathcal{A}$. Then $q_\beta^\star$ in (6) is the unique solution to*

$$q_\beta^\star = B_\beta q_\beta^\star, \quad where \quad q_\beta^\star = q^\star(\cdot, \cdot, \beta). \quad (8)$$

Although the Bellman operator defined in (8) can be used to compute the value function when the transition probabilities are known, it is inconvenient in the model-free reinforcement learning setting where the state-action value function must be estimated directly from samples.

We now turn to an alternative definition of the Bellman operator that can be used to estimate the value function directly from samples. To develop our ERM Q-learning algorithm, we need to define a Bellman operator that is amenable to computing its fixed point using stochastic gradient descent. For this purpose, we use the *elicitability* property of risk measures [7]. ERM is known to be elicitable using the following *loss* function $\ell_\beta \colon \mathbb{R} \to \mathbb{R}$:

$$\ell_\beta(z) := \beta^{-1}(\exp(-\beta \cdot z) - 1) + z, \quad \ell'_\beta(z) = 1 - \exp(-\beta \cdot z), \quad \ell''_\beta(z) = \beta \cdot \exp(-\beta \cdot z). \quad (9)$$

The functions $\ell'$ and $\ell''$ are the first and second derivatives, which are important in constructing and analyzing the Q-learning algorithm.

The following proposition summarizes the elicitability property of ERM, which we need to develop our Q-learning algorithm. Elicitable risk measures can be estimated using regression from samples in a model-free way.

**Proposition 3.2.** *For each $\tilde{x} \in \mathbb{X}$ and $\beta > 0$:*

$$\mathrm{ERM}_\beta[\tilde{x}] = \arg\min_{y \in \mathbb{R}} \mathbb{E}[\ell_\beta(\tilde{x} - y)], \quad (10)$$

*where the minimum is unique because $\ell_\beta$ is strictly convex.*

Using the elicitability property, we define the Bellman operator $\hat{B}_\beta \colon \mathcal{Q} \to \mathcal{Q}$ for $\beta \in \mathcal{B}$ as

$$(\hat{B}_\beta q)(s, a) := \arg\min_{y \in \mathbb{R}} \mathbb{E}^{a,s} \left[ \ell_\beta \left( r(s, a, \tilde{s}_1) + \max_{a' \in \mathcal{A}} q(\tilde{s}_1, a', \beta) - y \right) \right], \quad \forall s \in \mathcal{S}, a \in \mathcal{A}. \quad (11)$$

Following Proposition 3.2, we get the following equivalence of the Bellman operator, which implies that their fixed points coincide.

**Theorem 3.3.** *For each $\beta > 0$ and $q \in \mathcal{Q}$, we have that*

$$B_\beta q = \hat{B}_\beta q.$$

We can introduce the ERM Q-learning algorithm in Algorithm 1 using the results above. This algorithm adapts the standard Q-learning approach to the risk-averse setting. Intuitively, it works as follows. Each iteration $i$ processes a single transition sample to update the optimal state-action value function estimate $\tilde{q}_i$. The value function estimates are random because the samples are random. The value function estimates are updated following a stochastic gradient descent along the derivative of the loss function $\ell_\beta$ defined in (9). Each sample is used to simultaneously update the state-action value function for multiple values of $\beta \in \mathcal{B}$.

---

**Algorithm 1:** ERM-TRC Q-learning algorithm

**Input:** Risk levels $\mathcal{B} \subseteq \mathbb{R}_{++}$, samples: $(\tilde{s}_i, \tilde{a}_i, \tilde{s}'_i)$, step sizes $\tilde{\eta}_i, i \in \mathbb{N}$, bounds $z_{\min}, z_{\max}$
**Output:** Estimate state-action value function $\tilde{q}_i$

1   $\tilde{q}_0(s, a, \beta) \leftarrow 0, \quad \forall s \in \mathcal{S}, a \in \mathcal{A}$ ;
2   **for** $i \in \mathbb{N}, s \in \mathcal{S}, a \in \mathcal{A}, \beta \in \mathcal{B}$ **do**
3      **if** $s = \tilde{s}_i \wedge a = \tilde{a}_i$ **then**
4         $\tilde{z}_i(\beta) \leftarrow r(s, a, \tilde{s}'_i) + \max_{a' \in \mathcal{A}} \tilde{q}_i(\tilde{s}'_i, a', \beta) - \tilde{q}_i(s, a, \beta)$ ;
5         **if** $\neg(z_{\min} \leq \tilde{z}_i(\beta) \leq z_{\max})$ **then return** $\tilde{q}_{i+1}(s, a, \beta) = -\infty$;
6         $\tilde{q}_{i+1}(s, a, \beta) \leftarrow \tilde{q}_i(s, a, \beta) - \tilde{\eta}_i \cdot (\exp(-\beta \cdot \tilde{z}_i(\beta)) - 1)$ ;
7      **else** $\tilde{q}_{i+1}(s, a, \beta) \leftarrow \tilde{q}_i(s, a, \beta)$ ;

---

There are three main differences between Algorithm 1 and the standard Q-learning algorithm described in Section 2. First, standard Q-learning aims to maximize the expectation objective, and the proposed algorithm maximizes the ERM-TRC objective. Second, the state-action value function update follows a sequence of stochastic gradient steps, but it replaces the quadratic loss function with the exponential loss function $\ell_\beta$ in (9). Third, $q$ values in the proposed algorithm need an additional bounded condition that the TD residual $\tilde{z}_i(\beta)$ is in $[z_{\min}, z_{\max}]$.

*Remark* 1. The $q$ value can be unbounded for two main reasons. First, as the value of $\beta$ increases, the ERM value function does not increase, as shown in Lemma A.2, and can reach $-\infty$, as shown in Theorem A.1. Second, the agent has some probability of repeatedly receiving a reward for the same action, leading to an uncontrolled increase in the $q$ value for that action.

Note that if $\tilde{z}_i(\beta)$ is outside the $[z_{\min}, z_{\max}]$ range, it indicates that the value of the risk level $\beta$ is so large that $\tilde{q}$ is unbounded. This aligns with the conclusion that the value function may be unbounded for a large risk factor [35, 44]. Detecting unbounded values of $q$ caused by random chance is useful but is challenging and beyond the scope of this work.

*Remark* 2. We now discuss how to estimate $z_{\min}$ and $z_{\max}$ in Algorithm 1. Assume that the sequences $\{\tilde{\eta}_i\}_{i=0}^{\infty}$ and $\{(\tilde{s}_i, \tilde{a}_i, \tilde{s}_i')\}_{i=0}^{\infty}$ used in Algorithm 1, $\beta > 0$, we have

$$z_{\min}(\beta) = -2\max\{|c - \beta \cdot d|, |c|\} - \|r\|_{\infty}, \quad z_{\max}(\beta) = 2\max\{|c - \beta \cdot d|, |c|\} + \|r\|_{\infty}.$$

Where $\|r\|_{\infty} = \max_{s,s' \in \mathcal{S}, a \in \mathcal{A}} |r(s, a, s')|$. The constants $c$ and $d$ can be estimated by Algorithm 3 in the appendix. For more details on the derivation, we refer the interested reader to Appendix A.4.

Now we explain how to leverage the elicitability and monotonicity of ERM to design Algorithm 1 as a stochastic gradient descent. Let $b = (s, a, \beta), s \in \mathcal{S}, a \in \mathcal{A}, \beta \in \mathcal{B}$, and $\xi > 0$, and define operators $G$ and $H$ as

$$(Gq)(b) := \frac{\partial}{\partial y} \mathbb{E}^{a,s}\Big[\ell_\beta\Big(r(s, a, \tilde{s}_1) + \max_{a' \in \mathcal{A}} q(\tilde{s}_1, a', \beta) - y\Big)\Big] \mid_{y = q(s, a, \beta)}, \tag{12}$$

$$(G_{s'}q)(b) := \frac{\partial}{\partial y} \mathbb{E}\Big[\ell_\beta\Big(r(s, a, s') + \max_{a' \in \mathcal{A}} q(s', a', \beta) - y\Big)\Big] \mid_{y = q(s, a, \beta)} \tag{13}$$

The operation $(Gq)(b)$ can be interpreted as the average gradient step where $\tilde{s}_1$ is a random variable representing the next state. The operation $(G_{s'}q)(b)$ can be interpreted as the individual gradient step for a specific next state $s'$.

$$(Hq)(b) := q(b) - \xi \cdot (Gq)(b), \qquad (H_{s'}q)(b) := q(b) - \xi \cdot (G_{s'}q)(b). \tag{14}$$

The operation $(Hq)(b)$ can be interpreted as the q-update for a random variable $\tilde{s}_1$ representing the next state. $(H_{s'}q)(b)$ can be interpreted as the q-update a specific next state $s'$.

**Lemma 3.4.** *Let* $b = (s, a, \beta), s \in \mathcal{S}, a \in \mathcal{A}, \beta \in \mathcal{B}, 1 \in \mathbb{R}^n$ *is the vector with all components equal to 1, and* $g > 0$, *then the $H$ operator defined in* (14) *satisfies the monotonicity property such that*

$$x(b) \le y(b) \Rightarrow (Hx)(b) \le (Hy)(b) \tag{a}$$

$$(Hq^\star)(b) = q^\star(b) \tag{b}$$

$$(Hq)(b) - 1 \cdot g \le H(q(b) - 1 \cdot g) \le H(q(b) + 1 \cdot g) \le (Hq)(b) + 1 \cdot g. \tag{c}$$

The proof has two main steps. First, rewrite $(Hq)(b)$ in terms of the ERM Bellman operator in (11). Second, prove the monotonicity property of $(Hq)(b)$, which is necessary for the convergence analysis of Algorithm 1. Fix $\beta > 0$ and some $b = (s, a, \beta), s \in \mathcal{S}, a \in \mathcal{A}$, fix $q$ and define

$$f(y) = \mathbb{E}^{s,a}\Big[\ell_\beta(r(s, a, \tilde{s}_1) + \max_{a' \in \mathcal{A}} q(\tilde{s}_1, a', \beta) - y)\Big]. \tag{15}$$

The function $f$ is $\ell$-strongly convex with an $L$-Lipschitz-continuous gradient from Lemma 4.4. Let $y^\star = \arg\min_{y \in \mathbb{R}} f(y) = (\hat{B}q)(b)$ and $\exists l \in [1/L, 1/\ell]$ such that

$$(Hq)(b) = (q - \xi Gq)(b) \overset{(a)}{=} q(b) - \xi f'(q(b)) \overset{(b)}{=} (1 - \xi/l)q(b) + \xi/l \cdot y^\star$$
$$\overset{(c)}{=} (1 - \xi/l)q(b) + \xi/l \cdot (\hat{B}q)(b). \tag{16}$$

Step (a) follows by algebraic manipulation from the definitions in (12) and (15). Step (b) follows from Lemma 2.1. Step (c) follows from the fact that $\hat{B}$ is the ERM Bellman operator defined in (11) and $y^\star$ is the unique solution to (10).

## 3.2 EVaR Q-learning Algorithm

In this section, we adapt our ERM Q-learning algorithm to compute the optimal policy for the static EVaR-TRC objective in (3). As mentioned in Section 2, EVaR is preferable to ERM because it is coherent and closely approximates VaR and CVaR. The construction of an optimal EVaR policy follows the standard methodology proposed in prior work [19, 44]. We only briefly summarize it in this paper.

The main challenge in solving (3) is that EVaR is not dynamically consistent, and the supremum in (2) may not be attained. We show that the EVaR-TRC problem can be reduced to a sequence of ERM-TRC problems, similarly to the discounted case [19] and the undiscounted case [44]. We define the objective function $h \colon \Pi \times \mathbb{R}_{++} \to \bar{\mathbb{R}}$:

$$
\begin{aligned}
h(\pi, \beta) &:= \mathrm{ERM}_\beta^{\pi,\mu}\left[q^\pi(\tilde{s}_0, \tilde{a}_0, \beta)\right] + \beta^{-1}\log(\alpha) \\
&= \lim_{t\to\infty} \mathrm{ERM}_\beta^{\pi,\mu}\left[\sum_{k=0}^{t-1} r(\tilde{s}_k, \tilde{a}_k, \tilde{s}_{k+1})\right] + \beta^{-1}\log(\alpha).
\end{aligned}
\tag{17}
$$

Recall that $\tilde{s}_0$ is distributed according to the initial distribution $\mu$ and $\tilde{a}_0$ is the action distributed according to the policy $\pi$. The equality in the equation above follows directly from the definition of the value function in (6).

We now compute a $\delta$-optimal EVaR-TRC policy for any $\delta > 0$ by solving a sequence of ERM-TRC problems. Following prior work [44], we replace the supremum over continuous $\beta$ in the definition of EVaR in (2) with a maximum over a *finite* set $\mathcal{B}(\beta_0, \delta)$ of discretized $\beta$ values chosen as

$$
\mathcal{B}(\beta_0, \delta) := \{\beta_0, \beta_1, \ldots, \beta_K\},
\tag{18a}
$$

where $0 < \beta_0 < \beta_1 < \cdots < \beta_K$, and

$$
\beta_{k+1} := \frac{\beta_k \log\frac{1}{\alpha}}{\log\frac{1}{\alpha} - \beta_k\delta}, \quad \beta_K \geq \frac{\log\frac{1}{\alpha}}{\delta}, \quad K \geq \frac{\log(\theta)}{\log(1-\theta)},
\tag{18b}
$$

for $\theta := -\beta_0 \cdot \delta / \log(\alpha)$, and the minimal $K$ that satisfies the inequality. Note that the construction in (18) differs in the choice of $\beta_0$ from $\mathcal{B}(\beta_0, \delta)$ in [19].

The following theorem summarizes the fact that the EVaR-TRC problem reduces to a sequence of ERM-TRC optimization problems.

**Theorem 3.5** (Theorem 4.2 in [43]). *For any $\delta > 0$ and a sufficiently small $\beta_0 > 0$, let*

$$
(\pi^\star, \beta^\star) \in \operatorname*{arg\,max}_{(\pi,\beta)\in\Pi\times\mathcal{B}(\beta_0,\delta)} h(\pi, \beta).
$$

*Then, the limits below exist and satisfy that*

$$
\lim_{t\to\infty} \mathrm{EVaR}_\alpha^{\pi^\star,\mu}\left[\sum_{k=0}^{t-1} r(\tilde{s}_k, \tilde{a}_k, \tilde{s}_{k+1})\right] \geq \sup_{\pi\in\Pi} \limsup_{\substack{t\to\infty \\ \beta>0}} h(\pi, \beta) - \delta.
\tag{19}
$$

---

**Algorithm 2:** EVaR-TRC Q-learning algorithm

**Data:** desired precision $\delta > 0$, risk level $\alpha \in (0,1)$, initial $\beta_0 > 0$, samples: $(\tilde{s}_i, \tilde{a}_i, \tilde{s}_i'), i \in \mathbb{N}$
**Result:** $\delta$-optimal policy $\pi^\star \in \Pi$
1 Construct $\mathcal{B}(\beta_0, \delta)$ as described in (18) ;
2 Compute $(\pi_\beta^\star, h^\star(\beta))$ by Algorithm 1 for each $\beta \in \mathcal{B}(\beta_0, \delta)$ where $h^\star(\beta) = \max_{\pi\in\Pi} h(\pi, \beta)$;
3 Let $\beta^\star \in \arg\max_{\beta\in\mathcal{B}(\beta_0,\delta)} h^\star(\beta)$;
4 **return** $\pi_{\beta^\star}^\star$

---

Algorithm 2 summarizes the procedure for computing a $\delta$-optimal EVaR policy. Note that the value of $h(\pi, \beta)$ may be $-\infty$, indicating either that the ERM-TRC objective is unbounded for $\beta$ or that the Q-learning algorithm diverged by random chance.

*Remark* 3. Note that Algorithm 2 assumes a small $\beta_0$ as its input. The derivation of $\beta_0$ is shown in A.5. It is unclear whether obtaining a prior bound on $\beta_0$ without knowing the model is possible. We, therefore, employ the heuristic outlined in Algorithm 3 that estimates a lower bound $x_{\min}$ and an upper bound $x_{\max}$ on the random variable of returns. Then, we set $\beta_0 := 8\delta/(x_{\max}-x_{\min})^2$.

## 4 Convergence Analysis

This section presents our main convergence guarantees for the ERM-TRC Q-learning and EVaR-TRC Q-learning algorithms. The proofs for this section are deferred to Appendix B.

We require the following standard assumption to prove the convergence of the proposed Q-learning algorithms. The intuition of Assumption 4.1 is that each state-action pair must be visited infinitely often. Note that when an individual episode terminates upon reaching the sink state, the agent restarts a new episode from a random initial state.

**Assumption 4.1.** The input to Algorithm 1 and Algorithm 2 satisfies that

$$\mathbb{P}[\tilde{s}_i' = s' \mid \mathcal{G}_{i-1}, \tilde{s}_i, \tilde{a}_i, \tilde{\eta}_i] = p(\tilde{s}_i, \tilde{a}_i, s'), \qquad \forall s' \in \mathcal{S}, \forall i \in \mathbb{N},$$

almost surely, where $\mathcal{G}_{i-1} := (\tilde{\eta}_l, (\tilde{s}_l, \tilde{a}_l, \tilde{s}_l'))_{l=0}^{i-1}$.

The following theorem shows that the proposed ERM-TRC Q-learning algorithm enjoys convergence guarantees comparable to standard Q-learning. As Remark 2 states, $z_{\min}$ and $z_{\max}$ are chosen to ensure that $q$ values are bounded.

**Theorem 4.2.** *For $\beta \in \mathcal{B}$, assume that the sequence $(\tilde{\eta}_i)_{i=0}^{\infty}$ and $(\tilde{s}_i, \tilde{a}_i, \tilde{s}_i')_{i=0}^{\infty}$ used in Algorithm 1 satisfies Assumption 4.1 and step size condition*

$$\sum_{i=0}^{\infty} \tilde{\eta}_i = \infty, \quad \sum_{i=0}^{\infty} \tilde{\eta}_i^2 < \infty,$$

*where $i \in \{i \in \mathbb{N} \mid (\tilde{s}_i, \tilde{a}_i) = (s, a)\}$, if $\tilde{z}_i \in [z_{\min}, z_{\max}]$ almost surely, then the sequence $(\tilde{q}_i)_{i=0}^{\infty}$ produced by Algorithm 1 convergences almost surely to $q_{\infty}$ such that $q_{\infty} = \hat{B}_{\beta} q_{\infty}$.*

The proof of Theorem 4.2 follows an approach similar to that in the proofs of standard Q-learning [8] with three main differences. First, the algorithm converges for an undiscounted MDP, and $\hat{B}_{\beta}$ is not a contraction captured by [8, assumption 4.4], which is restated as Assumption B.2. Instead, we show that $\hat{B}_{\beta}$ satisfies monotonicity in Lemma 4.3. Second, Assumption B.2 leads to a convergence result somewhat weaker than the results for a contraction. Then, a separate boundedness condition [8, proposition 4.6] is imposed. Third, using the exponential loss function $\ell_{\beta}$ in (9) requires a more careful choice of step-size than the standard analysis.

**Lemma 4.3.** *For $\beta \in \mathcal{B}$, let $1 \in \mathbb{R}^n$ represent the vector of all ones. Then the ERM Bellman operator defined in (11) satisfies the monotonicity property if for some $g > 0$ and $q^{\star} \colon \mathbb{R}^{\mathcal{S} \times \mathcal{A}} \to \mathbb{R}^{\mathcal{S} \times \mathcal{A}}$ and for all $q \colon \mathbb{R}^{\mathcal{S} \times \mathcal{A}} \to \mathbb{R}^{\mathcal{S} \times \mathcal{A}}$ it satisfies that*

$$x \leq y \quad \Rightarrow \quad \hat{B}_{\beta} x \leq \hat{B}_{\beta} y \tag{a}$$

$$\hat{B}_{\beta} q^{\star} \quad = \quad q^{\star} \tag{b}$$

$$\hat{B}_{\beta} q - g \cdot 1 \quad \leq \quad \hat{B}_{\beta}(q - g \cdot 1) \quad \leq \quad \hat{B}_{\beta}(q + g \cdot 1) \quad \leq \quad \hat{B}_{\beta} q + g \cdot 1 \tag{c}$$

*Here, the relations hold element-wise.*

The following lemma shows that the loss function $\ell_{\beta}$ in (9) is strongly convex and has a Lipschitz-continuous gradient. These properties are instrumental in showing the uniqueness of our value functions and analyzing the convergence of our Q-learning algorithms by analyzing them as a form of stochastic gradient descent.

**Lemma 4.4.** *The function $\ell_{\beta} \colon [z_{\min}, z_{\max}] \to \mathbb{R}$ defined in (9) is $l$-strongly convex with $l = \beta \exp(-\beta \cdot z_{\max})$ and its derivative $\ell_{\beta}'(z)$ is $L$-Lipschitz-continuous with $L = \beta \cdot \exp(-\beta \cdot z_{\min})$.*

Corollary 4.5, which follows immediately from Theorem 3.5, shows that Algorithm 2 enjoys convergence guarantees and converges to a $\delta$-optimal EVaR policy.

**Corollary 4.5.** *For $\alpha \in (0, 1)$ and $\delta > 0$, assume that the sequence $(\tilde{\eta}_i)_{i=0}^{\infty}$ and $(\tilde{s}_i, \tilde{a}_i, \tilde{s}_i')_{i=0}^{\infty}$ used in Algorithm 1 satisfies Assumption 4.1 and step size condition*

$$\sum_{i=0}^{\infty} \tilde{\eta}_i = \infty, \quad \sum_{i=0}^{\infty} \tilde{\eta}_i^2 < \infty,$$

*where $i \in \{i \in \mathbb{N} \mid (\tilde{s}_i, \tilde{a}_i) = (s, a)\}$, if $\tilde{z}_i \in [z_{\min}, z_{\max}]$. Then Algorithm 2 converges to the $\delta$-optimal stationary policy $\pi^{\star}$ for the EVaR-TRC objective in (3) almost surely.*

The proof of Corollary 4.5 follows a similar approach to the proofs of Theorem 4.2. However, we also need to show that one can obtain an optimal ERM policy for an appropriately chosen $\beta$ that approximates an optimal EVaR policy arbitrarily closely.

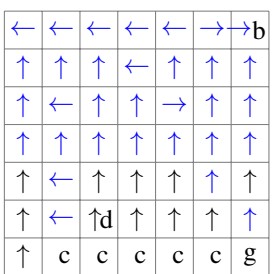

Figure 1: EVaR optimal policy with $\alpha = 0.2$

Figure 2: EVaR optimal policy with $\alpha = 0.6$

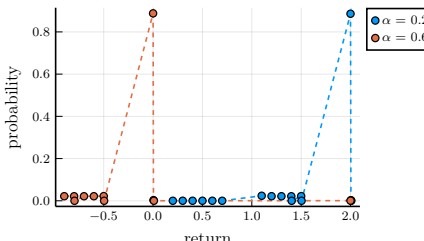

Figure 3: Comparison of return distributions of two optimal EVaR policies.

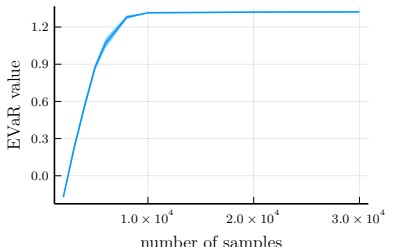

Figure 4: Mean and standard deviation of EVaR values with $\alpha = 0.2$ on CW domain

## 5   Numerical Evaluation

In this section, we evaluate our algorithms on two tabular domains: cliff walking (CW) [4] and gambler's ruin (GR) [44]. The source code is available at `https://github.com/suxh2019/ERM_EVaR_Q`. First, we evaluate Algorithm 2 on the CW domain. In this problem, an agent starts with a random state (cell in the grid world) that is uniformly distributed over all non-sink states and walks toward the goal state labeled by $g$ shown in Figure 1. At each step, the agent takes one of four actions: up, down, left, or right. The action will be performed with a probability of $0.91$, and the other three actions will also be performed separately with a probability of $0.03$. If the agent hits the wall, it will stay in its place. The reward is zero in all transitions except for the states marked with $c$, $d$, and $g$. The agent receives 2 in state $g$, $0.004$ in state $d$, $-0.5, -0.6, -0.7, -0.8$ and $-0.9$ in cliff region marked with $c$ from the left to the right. If an agent steps into the cliff region, it will immediately be returned to the state marked with $b$. Note that the agent has unlimited steps, and the future rewards are not discounted.

We use Algorithm 3 to compute $c$ and $d$ in Remark 2 and estimate the bounds of $\tilde{z}_i(\beta)$ in Algorithm 1. Figure 1 and Figure 2 show the optimal policies for EVaR with risk level $\alpha = 0.2$ and $\alpha = 0.6$ separately. Since the optimal policy is stationary, it can be analyzed visually. The arrow direction indicates the optimal action to take in that state. Note that the optimal actions for states $c$ and $g$ are the same and are omitted. In Figure 1, the agent moves right to the last column of the grid world and then moves down to reach the goal. When the agent is close to the cliff region, it moves up to avoid risk. In Figure 2, different optimal actions are highlighted in blue. As we can see, for different risk levels $\alpha$, when the agent is near the cliff region, it exhibits consistent behaviors to avoid falling off the cliff. For the remaining states, the agent behaves differently.

Second, to understand the impact of risk aversion on the structure of returns, we simulate the two optimal EVaR policies over $48,000$ episodes and display the distribution of returns in Figure 3. The $x$-axis represents the possible return the agent can get for each episode, and the $y$-axis represents the probability of getting a certain amount of return. For each episode, the agent takes $20,000$ steps and collects all rewards during the path. When $\alpha = 0.6$, the return is in $(-1, 2]$, and its mean value is $-0.074$, and its standard deviation is $0.228$. When $\alpha = 0.2$, the return is in $(0, 2]$, its mean value is $1.92$, and its standard deviation is $0.228$. Overall, Figure 3 shows that for the lower value of $\alpha$, the agent has a higher probability of avoiding falling off the cliff.

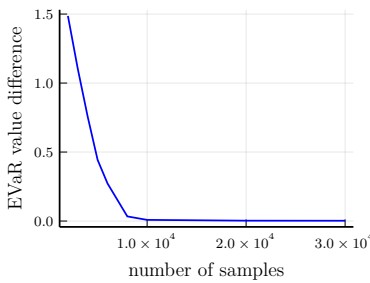
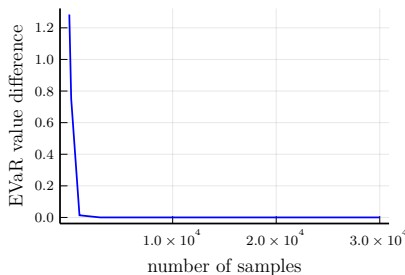

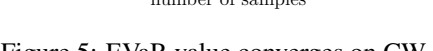

Figure 5: EVaR value converges on CW          Figure 6: EVaR value converges on GR

Third, to assess the stability of Algorithm 2, we use six random seeds to generate samples, compute the optimal policies, and calculate the EVaR values on the CW domain. In Figure 4, the risk level $\alpha$ is 0.2, the $x$ axis represents the number of samples, and the $y$ axis represents the mean value and standard deviation of the EVaR values. The standard deviation is 0.015, and our Q-learning algorithm converges to similar solutions across different numbers of samples. Note that our Q-learning algorithm could converge to a different optimal EVaR policy depending on the learning rate and the number of samples.

Finally, we evaluate the convergence of Algorithm 2, which approximates the EVaR value calculated from the linear programming(LP) [44] on CW and GR domains. The $x$ axis represents the number of samples, the $y$ axis represents the difference in EVaR values computed by LP and Q-learning algorithms, and the risk level $\alpha$ is 0.2. Figure 5 and Figure 6 show that the EVaR value difference decreases to zero when the number of samples is around 20,000 on the CW and GR domains. We illustrate the convergence results for $\alpha = 0.2$, but the conclusion applies to any $\alpha$ value. Overall, our Q-learning algorithm converges to the optimal value function.

## 6   Conclusion and Limitations

In this paper, we proposed two new risk-averse model-free algorithms for risk-averse reinforcement learning objectives. We also proved the convergence of the proposed algorithm under mild assumptions. To the best of our knowledge, these are the first risk-averse model-free algorithms for the TRC criterion. Studying the TRC criterion in risk-averse RL is essential because it has dramatically different properties in risk-averse objectives than the discounted criterion.

An important limitation of this work is that our algorithms focus on the tabular setting and do not analyze the impact of value function approximation. However, the proposed algorithm is not limited to tabular representations. It is a general model-free gradient-based method that can be combined with any differentiable value function approximators. This work lays out solid foundations for building scalable approximation reinforcement learning algorithms. In particular, a strength of Q-learning is that it can be coupled with general value function approximation schemes, such as deep neural networks. Although extending convergence guarantees to such approximate settings is challenging, our convergence results show that our Q-learning algorithm is a sound foundation for scalable, practical algorithms.

Our theoretical analysis has two main limitations that may preclude its use in some practical settings. First, we must choose the limits $z_{\min}, z_{\max}$. If these limits are set too small, the algorithms may fail to compute a good solution, and if they are too large, the algorithms may be excessively slow to detect the divergence of the TRC criterion for large values of $\beta$. It may be possible to use existing TD algorithms to improve how quickly we detect the divergence. The second limitation is the need to select the parameter $\beta_0$ that guarantees the existence of a $\delta$-optimal EVaR solution. A natural choice for $\beta_0$, proposed in [44], requires several runs of Q-learning just to establish an appropriate $\beta_0$.

## Acknowledgments

We thank the anonymous reviewers for their detailed reviews and thoughtful comments, which significantly improved the paper's clarity. This work was supported, in part, by NSF grants 2144601 and 2218063.

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

# A Proofs of Section 3

## A.1 Standard Results

For the model-based approach, Theorem A.1 shows that for an infinite horizon, the optimal exponential value function $w^{\infty,\star}$ is attained by a stationary deterministic policy and is a fixed point of the exponential Bellman operator. Following this theorem, we show that the ERM state-action value function can be computed as the fixed point of the Bellman operator shown in Theorem 3.1 in the main paper.

**Theorem A.1** ( [44], Theorem 3.3). *Whenever $w^{\infty,\star} > -\infty$ there exists $\pi^\star = (d^\star)_\infty \in \Pi_{SD}$ such that*

$$w^{\infty,\star} = w^\infty(\pi^\star) = L^{d^\star} w^{\infty,\star},$$

*and $w^{\infty,\star}$ is the unique value that satisfies this equation.*

**Lemma A.2** ( [18], Lemma A.7). *The function $\beta \mapsto \mathrm{ERM}_\beta[X]$ for any random variable $X \in \mathbb{X}$ and $\beta > 0$ is continuous and non-increasing*

**Lemma A.3** ( [18], Lemma A.8). *Let $X \in \mathbb{X}$ be a bounded random variable such that $x_{min} < X < x_{max}$ a.s. Then, for any risk level $\beta > 0$, $\mathrm{ERM}_\beta[\cdot]$ can be bounded as*

$$\mathbb{E}[X] - \frac{\beta(x_{max} - x_{min})^2}{8} \leq \mathrm{ERM}_\beta[X] \leq \mathbb{E}[X]$$

## A.2 Proof of Theorem 3.1

*Proof.* The exponential value function $w^\star(s)$ is the unique solution to the exponential Bellman equation [44, theorem 3.3]. Therefore, for a state $s \in \mathcal{S}$ and the optimal policy $\pi$, state value function $v^\pi(s)$, and the exponential state value function $w^\pi(s)$, $\beta \in \mathcal{B}$, we have that

$$v^\pi(s) = -\beta^{-1} \log(-w^\pi(s)) \tag{20}$$

Because of the bijection between $w^\star(s)$ and $v^\star(s)$, we can conclude that $v^\star(s)$ is the unique solution to the regular Bellman equation.

The optimal state value function can be rewritten in terms of the optimal state-action value function in (21)

$$v^\star(s) = \max_{a \in \mathcal{A}} q^\star(s, a, \beta) \tag{21}$$

Then $q^\star(s, a, \beta)$ exists. The value $q^\star(s, a, \beta)$ is unique directly from the uniqueness of $v^\star$.

$\square$

## A.3 Proof of Proposition 3.2

*Proof.* The proof is broken into two parts. First, we show that $\ell_\beta(\tilde{x} - y)$ is strictly convex, and a minimum value of $\ell_\beta$ exists. Second, we show that the minimizer of $\mathbb{E}[\ell_\beta(\tilde{x} - y)]$ is equal to $\mathrm{ERM}_\beta[\tilde{x}]$.

Take the first derivative of $\ell_\beta(\tilde{x} - y)$ with respect to $y$.

$$\begin{aligned}
\frac{\partial \ell_\beta(\tilde{x} - y)}{\partial y} &= \frac{\partial \left(\beta^{-1}(e^{-\beta \cdot (\tilde{x}-y)} - 1) + (\tilde{x} - y)\right)}{\partial y} \\
\ell'_\beta(y) &= \beta^{-1}(e^{-\beta \cdot (\tilde{x}-y)}) \cdot \beta - 1 \\
\ell'_\beta(y) &= e^{-\beta \cdot (\tilde{x}-y)} - 1
\end{aligned} \tag{22}$$

Take the second derivative of $\ell_\beta(\tilde{x} - y)$ with respect to $y$

$$\begin{aligned}
\frac{\partial^2 \ell_\beta}{\partial y^2} &= \frac{\partial(e^{-\beta \cdot (\tilde{x}-y)} - 1)}{\partial y} \\
\ell''_\beta(y) &= e^{-\beta \cdot (\tilde{x}-y)} \cdot \beta \\
\ell''_\beta(y) &= \beta e^{-\beta \cdot (\tilde{x}-y)} > 0
\end{aligned} \tag{23}$$

Therefore, $\ell_\beta$ is strongly convex with respect to $y$, and the minimum value of $\ell_\beta$ exists.

Second, take the first derivative of $\mathbb{E}[\ell_\beta(\tilde{x}-y)]$ with respective to $y$

$$\frac{\partial \mathbb{E}\big[\beta^{-1}(e^{-\beta\cdot(\tilde{x}-y)}-1)+(\tilde{x}-y)\big]}{\partial y} = 0$$

$$\mathbb{E}\big[e^{-\beta\cdot(\tilde{x}-y)}-1\big] = 0$$

$$\mathbb{E}\big[e^{-\beta\cdot(\tilde{x}-y)}\big] = 1$$

$$\mathbb{E}\big[e^{-\beta\cdot\tilde{x}}\big]\cdot\mathbb{E}\big[e^{\beta\cdot y}\big] = 1$$

$$\mathbb{E}\big[e^{-\beta\cdot\tilde{x}}\big]\cdot e^{\beta\cdot y} = 1$$

$$e^{-\beta\cdot y} = \mathbb{E}\big[e^{-\beta\cdot\tilde{x}}\big]$$

$$y = -\beta^{-1}\log(\mathbb{E}\big[e^{-\beta\cdot\tilde{x}}\big])$$

Therefore, the minimizer of $\mathbb{E}[\ell_\beta(\tilde{x}-y)]$ is equal to $\mathrm{ERM}_\beta[\tilde{x}]$. $\qquad\square$

### A.4 Derivation of Remark 2

---
**Algorithm 3:** A heuristic algorithm for computing $z$ bounds

---
**Input:** Risk levels $\beta_c = 1^{-10}$, samples: $(\tilde{s}_i, \tilde{a}_i, \tilde{s}'_i)$, step sizes $\tilde{\eta}_i, i \in \mathbb{N}$
**Output:** Estimated expectation value $c$, $x_{\min}$, and $x_{\max}$
1   $\tilde{q}_0(s,a,\beta_c) \leftarrow 0, \tilde{x}(s,a,\beta_c) \leftarrow 0, \quad \forall s \in \mathcal{S}, a \in \mathcal{A}$ ;
2   **for** $i \in \mathbb{N}, s \in \mathcal{S}, a \in \mathcal{A}$ **do**
3     **if** $s = \tilde{s}_i \wedge a = \tilde{a}_i$ **then**
4       $\tilde{z}_i(\beta_c) \leftarrow r(s,a,\tilde{s}'_i) + \max_{a'\in\mathcal{A}} \tilde{q}_i(\tilde{s}'_i, a', \beta_c) - \tilde{q}_i(s,a,\beta_c)$ ;
5       $\tilde{q}_{i+1}(s,a,\beta_c) \leftarrow \tilde{q}_i(s,a,\beta_c) - \tilde{\eta}_i \cdot (\exp(-\beta\cdot\tilde{z}_i(\beta_c))-1)$ ;
6       $\tilde{x}(s,a,\beta_c) \leftarrow \tilde{x}(s,a,\beta_c) + r(s,a)$
7     **else** $\tilde{q}_{i+1}(s,a,\beta_c) \leftarrow \tilde{q}_i(s,a,\beta_c)$ ;
8   $c \leftarrow \max \tilde{q}_\infty(s,a,\beta_c), s \in \mathcal{S}, a \in \mathcal{A}$ ;
9   $x_{\min} \leftarrow \min \tilde{x}(s,a,\beta_c), s \in \mathcal{S}, a \in \mathcal{A}$ ;
10   $x_{\max} \leftarrow \max \tilde{x}(s,a,\beta_c), s \in \mathcal{S}, a \in \mathcal{A}$ ;
11   $d \leftarrow (x_{\max} - x_{\min})^2/8$.

---

From line 4 of Algorithm 1, we have

$$\tilde{z}_i(\beta) \leftarrow r(s,a,\tilde{s}'_i) + \max_{a'\in\mathcal{A}} \tilde{q}_i(\tilde{s}'_i, a', \beta) - \tilde{q}_i(s,a,\beta), s \in \mathcal{S}, a \in \mathcal{A}, \beta \in \mathcal{B}, i \in \mathbb{N}$$

Do some algebraic manipulation,

$$| \tilde{z}_i(\beta) | \leq \|r\|_\infty + |q|_{\max} + |q|_{\max}$$
$$| \tilde{z}_i(\beta) | \leq \|r\|_\infty + 2|q|_{\max}$$

where $\|r\|_\infty = \max\limits_{s,s'\in\mathcal{S},a\in\mathcal{A}} |r(s,a,s')|$ and $|q|_{\max} = \max\limits_{s\in\mathcal{S},a\in\mathcal{A}} |\tilde{q}_i(s,a,\beta)|, i \in \mathbb{N}$.

Let us estimate $|q|_{\max}$. For any random variable $\tilde{x}$, for any risk level $\beta \in \mathcal{B}$, $\mathrm{ERM}_\beta[\tilde{x}]$ is non-increasing in Lemma A.2 and satisfies monotonicity in Lemma A.3,

$$\mathbb{E}[\tilde{x}] - \beta(x_{\max} - x_{\min})^2/8 \leq \mathrm{ERM}_\beta[\tilde{x}] \leq \mathbb{E}[\tilde{x}]$$

That is,

$$\mathbb{E}[\tilde{x}] - \beta(x_{\max} - x_{\min})^2/8 \leq \tilde{q}_\infty(s,a,\beta) \leq \mathbb{E}[\tilde{x}], s \in \mathcal{S}, a \in \mathcal{A}$$

Then, for any $\beta \in \mathcal{B}$,

$$|q|_{\max} \approx \max_{s\in\mathcal{S},a\in\mathcal{A}} |\tilde{q}_\infty(s,a,\beta)|$$
$$= \max\{|\mathbb{E}[\tilde{x}] - \beta(x_{\max} - x_{\min})^2/8|, |\mathbb{E}[\tilde{x}]|\}$$

As mentioned in Section 2,

$$\mathrm{ERM}_0[\tilde{x}] = \lim_{\beta \to 0^+} \mathrm{ERM}_\beta[\tilde{x}] = \mathbb{E}[\tilde{x}]$$

Then

$$\mathbb{E}[\tilde{x}] \approx \max_{s \in \mathcal{S}, a \in \mathcal{A}} \tilde{q}_\infty(s, a, 0^+)$$

We estimate $\mathbb{E}[\tilde{x}]$, $x_{\max}$, $x_{\min}$ by setting $\beta_c = 1^{-10}$ in Algorithm 3. Therefore, for any $\beta \in \mathcal{B}$,

$$z_{\min}(\beta) = -||r||_\infty - 2\max\{|c - \beta \cdot d|, |c|\}, \quad z_{\max}(\beta) = ||r||_\infty + 2\max\{|c - \beta \cdot d|, |c|\}$$

where $c = \max \tilde{q}_\infty(s, a, \beta_c), s \in \mathcal{S}, a \in \mathcal{A}$ and $d = (x_{\max} - x_{\min})^2/8$.

### A.5 $\beta_0$ Derivation

Let us derive $\beta_0$. For a desired precision $\delta > 0$, $\beta_0$ is chosen such that

$$\mathbb{E}^{\pi^\star, \mu}[\tilde{x}] - \mathrm{ERM}_{\beta_0}^{\pi^\star, \mu}[\tilde{x}] \le \delta \tag{24}$$

From Lemma A.3, for a random variable $\tilde{x}$ and $\beta > 0$, we have

$$\mathbb{E}[\tilde{x}] - \mathrm{ERM}_\beta[\tilde{x}] \le \beta \cdot (x_{\max} - x_{\min})^2/8$$

Then we have

$$\mathbb{E}^{\pi^\star, \mu}[\tilde{x}] - \mathrm{ERM}_{\beta_0}^{\pi^\star, \mu}[\tilde{x}] \le \beta_0 \cdot (x_{\max} - x_{\min})^2/8 \tag{25}$$

When the equality conditions in (24) and (25) hold, we have

$$\beta_0 \cdot (x_{\max} - x_{\min})^2/8 = \delta$$

$$\beta_0 = \frac{8\delta}{(x_{\max} - x_{\min})^2}$$

where $x_{\max}$ and $x_{\min}$ are estimated in Algorithm 3.

## B  Proofs of Section 4

### B.1  Standard Convergence Results

Our convergence analysis of Q-learning algorithms is guided by the framework [8, section 4]. We summarize the framework in this section. Consider the following iteration for some random sequence $\tilde{r}_i : \Omega \to \mathbb{R}^\mathcal{N}$ where $\mathcal{N} = \{1, 2, \cdots, n\}$, then Q-learning is defined as

$$\begin{aligned} \tilde{r}_{i+1}(b) &= (1 - \tilde{\theta}_i(b)) \cdot \tilde{r}_i(b) + \tilde{\theta}_i(b) \cdot ((H\tilde{r}_i)(b) + \tilde{\phi}_i(b)), \quad i = 0, 1, \cdots \\ &= \tilde{r}_i(b) + \tilde{\theta}_i(b) \cdot ((H\tilde{r}_i)(b) + \tilde{\phi}_i(b) - \tilde{r}_i(b)), \quad i = 0, 1, \cdots \end{aligned} \tag{26}$$

for all $b \in \mathcal{N}$, where $H : \mathbb{R}^\mathcal{N} \to \mathbb{R}^\mathcal{N}$ is some possibly non-linear operator, $\tilde{\theta}_i : \Omega \to \mathbb{R}_{++}$ is a step size, and $\tilde{\phi}_i : \Omega \to \mathbb{R}^\mathcal{N}$ is some random noise sequence. The random history $\mathcal{F}_i$ at iteration $i = 1, \cdots$ is denoted by

$$\mathcal{F}_i = (\tilde{r}_0, \cdots, \tilde{r}_i, \tilde{\phi}_0, \cdots, \tilde{\phi}_{i-1}, \tilde{\theta}_0, \cdots, \tilde{\theta}_i).$$

The following assumptions will be needed to analyze and prove the convergence of our algorithms.

**Assumption B.1.** [assumption 4.3 in [8]]

(a) For every $i$ and $b$, we have

$$\mathbb{E}[\tilde{\phi}(b)|\mathcal{F}_i] = 0$$

(b) There exists a norm $|| \cdot ||$ on $\mathbb{R}^\mathcal{N}$ and constants $A$ and $B$ such that

$$\mathbb{E}[\tilde{\phi}_i(b)^2|\mathcal{F}_i] \le A + B||\tilde{r}_i||^2$$

**Assumption B.2.** [assumption 4.4 in [8]]

(a) The mapping $H$ is monotone: that is, if $r \leq \bar{r}$, then $Hr \leq H\bar{r}$.

(b) There exists a unique vector $r^\star$ satisfying $Hr^\star = r^\star$.

(c) If $e \in \mathbb{R}^n$ is the vector with all components equal to 1, and if $\eta$ is a positive scalar, then

$$Hr - \eta \cdot e \leq H(r - \eta \cdot e) \leq H(r + \eta \cdot e) \leq Hr + \eta \cdot e$$

Note that Assumption B.2 leads to a convergence result somewhat weaker than the results for weighted maximum norm pseudo-contraction. Then we need a separate boundedness condition [8, proposition 4.6], which is restated here as Proposition B.3.

**Proposition B.3** (proposition 4.6 in [8]). *Let $r_t$ be the sequence generated by the iteration shown in Equation (26). We assume that the bth component $\tilde{r}(b)$ of $\tilde{r}$ is updated according to Equation (26), with the understanding that $\tilde{\theta}_i(b) = 0$ if $\tilde{r}(b)$ is not updated at iteration $i$. $\tilde{\phi}_i(b)$ is a random noise term. Then we assume the following:*

*(a) The step sizes $\tilde{\theta}_i(b)$ are nonnegative and satisfy*

$$\sum_{i=0}^{\infty} \tilde{\theta}_i(b) = \infty, \quad \sum_{i=0}^{\infty} \tilde{\theta}_i^2(b) < \infty,$$

*(b) The noise terms $\tilde{\phi}_i(b)$ satisfies Assumption B.1,*

*(c) The mapping $H$ satisfies Assumption B.2.*

*If the sequence $\tilde{r}_i$ is bounded with probability 1, then $\tilde{r}_i$ converges to $r^\star$ with probability 1.*

## B.2 Proof of Theorem 4.2

### B.2.1 Operator Definitions

Let $b = (s, a, \beta), s \in \mathcal{S}, a \in \mathcal{A}, \beta \in \mathcal{B}$, and $\xi > 0$, we use some operators $G$ defined in (12) and (13), and $H$ defined in (14) for the convergence analysis of Algorithm 1 and Algorithm 2.

$$(Gq)(b) := \frac{\partial}{\partial y} \mathbb{E}^{a,s}\Big[\ell_\beta\Big(r(s,a,\tilde{s}_1) + \max_{a' \in \mathcal{A}} q(\tilde{s}_1, a', \beta) - y\Big)\Big]\Big|_{y=q(s,a,\beta)}$$

$$= \mathbb{E}^{a,s}\Big[\partial\ell_\beta\Big(r(s,a,\tilde{s}_1) + \max_{a' \in \mathcal{A}} q(\tilde{s}_1, a', \beta) - q(s,a,\beta)\Big)\Big]$$

$\tilde{s}_1$ is a random variable representing the next state. When $\tilde{s}_1$ is used in an expectation with a superscript, such as $\mathbb{E}^{a,s}$, then it does not represent a sample $\tilde{s}_i$ with $i = 1$. Still, instead it represents the transition from $\tilde{s}_0$ to $\tilde{s}_1$ distributed as $p(s, a, \cdot)$.

$$(G_{s'}q)(b) := \frac{\partial}{\partial y} \mathbb{E}\Big[\ell_\beta\Big(r(s,a,s') + \max_{a' \in \mathcal{A}} q(s', a', \beta) - y\Big)\Big]\Big|_{y=q(s,a,\beta)}$$

$$= \mathbb{E}\Big[\partial\ell_\beta\Big(r(s,a,s') + \max_{a' \in \mathcal{A}} q(s', a', \beta) - q(s,a,\beta)\Big)\Big]$$

$$(Hq)(b) := q(b) - \xi \cdot (Gq)(b), \qquad (H_{s'}q)(b) := q(b) - \xi \cdot (G_{s'}q)(b)$$

Consider a random sequence of inputs $(\tilde{s}_i, \tilde{a}_i)_{i=0}^{\infty}$ in Algorithm 1. We can define a real-valued random variable $\tilde{\phi}(s, a)$ for $s \in \mathcal{S}, a \in \mathcal{A}, \beta > 0$ in (27).

$$\tilde{\phi}_i(s, a, \beta) = \begin{cases} (H_{\tilde{s}_i'}\tilde{q}_i)(s,a) - (H\tilde{q}_i)(s,a), & \text{if } (\tilde{s}_i, \tilde{a}_i) = (s, a) \\ 0, & \text{otherwise} \end{cases} \tag{27}$$

$\tilde{\phi}_i$ can be interpreted as the random noise.

$$\tilde{\theta}_i(s, a, \beta) = \begin{cases} \tilde{\eta}_i/\xi, & \text{if } (\tilde{s}_i, \tilde{a}_i) = (s, a) \\ 0, & \text{otherwise} \end{cases} \tag{28}$$

We denote by $\mathcal{F}_i$ the history of the algorithm until step $i$, which can be defined as

$$\mathcal{F}_i = \{\tilde{q}_0, \cdots, \tilde{q}_i, \tilde{\phi}_0, \cdots, \tilde{\phi}_{i-1}, \tilde{\theta}_0, \cdots, \tilde{\theta}_i\} \tag{29}$$

**Lemma B.4.** *The random sequences of iterations followed by Algorithm 1 satisfies*

$$\tilde{q}_{i+1}(s,a,\beta) = \tilde{q}_i(s,a,\beta) + \tilde{\theta}_i(s,a,\beta) \cdot (H\tilde{q}_i + \tilde{\phi}_i - \tilde{q}_i)(s,a,\beta), \forall s \in \mathcal{S}, a \in \mathcal{A}, \beta \in \mathcal{B}, i \in \mathbb{N}, a.s.,$$

*where the terms are defined in Equations* (12) *to* (14)*,* (27) *and* (28)*.*

*Proof.* We prove this claim by induction on $i$. The base case holds immediately from the definition. To prove the inductive case, suppose that $i \in \mathbb{N}$ and we prove the result in the following two cases.

Case 1: suppose that $\tilde{b} = (\tilde{s}, \tilde{a}, \beta) = (s, a, \beta) = b$, then by algebraic manipulation

$$
\begin{aligned}
\tilde{q}_{i+1}(b) &= \tilde{q}_i(b) + \tilde{\theta}_i(b)((H\tilde{q}_i)(b) + \tilde{\phi}_i(b) - \tilde{q}_i(b)) \\
&= \tilde{q}_i(b) + \tilde{\theta}_i(b)((H\tilde{q}_i)(b) + (H_{\tilde{s}'_i}\tilde{q}_i)(b) - (H\tilde{q}_i)(b) - \tilde{q}_i(b)) \\
&= \tilde{q}_i(b) + \tilde{\theta}_i(b)((H_{\tilde{s}'_i}\tilde{q}_i)(b) - \tilde{q}_i(b)) \\
&= \tilde{q}_i(b) + \tilde{\theta}_i(b)((\tilde{q}_i - \xi G_{\tilde{s}'_i}\tilde{q}_i)(b) - \tilde{q}_i(b)) \\
&= \tilde{q}_i(b) - \tilde{\eta}_i((G_{\tilde{s}'_i}\tilde{q}_i)(b)) \\
&= \tilde{q}_i(b) - \tilde{\eta}_i(\partial \ell_\beta(r(s,a,\tilde{s}'_i) + \max_{a' \in \mathcal{A}} \tilde{q}_i(\tilde{s}'_i, a', \beta) - \tilde{q}_i(b)))
\end{aligned}
$$

Case 2: suppose that $\tilde{b} = (\tilde{s}, \tilde{a}, \beta) \neq (s, a, \beta) = b$, then by algebraic manipulation, the algorithm does not change the $q$:

$$
\begin{aligned}
\tilde{q}_{i+1}(b) &= \tilde{q}_i(b) + \tilde{\theta}_i(b)((H\tilde{q}_i)(b) + \tilde{\phi}_i(b) - \tilde{q}_i(b)) \\
&= \tilde{q}_i(b) + 0 \cdot ((H\tilde{q}_i)(b) + \tilde{\phi}_i(b) - \tilde{q}_i(b)) \\
&= \tilde{q}_i(b)
\end{aligned}
$$

$\square$

### B.2.2 Random Noise Analysis

We restate Lemma C.16 in [17] here as Lemma B.5, which is useful for proving properties of the random noise.

**Lemma B.5** (Lemma C.16 in [17])**.** *Under Assumption 4.1:*

$$\mathbb{P}[\tilde{s}'_i = s' | \mathcal{G}_{i-1}, \tilde{b}_i, \tilde{\xi}_i, \mathcal{F}_i] = p(\tilde{s}_i, \tilde{a}_i, s'), a.s.,$$

*for each $s' \in \mathcal{S}$ and $i \in \mathbb{N}$.*

**Lemma B.6.** *The noise $\tilde{\phi}_i$ defined in* (27) *satisfies*

$$\mathbb{E}[\tilde{\phi}_i(s,a,\beta) | \mathcal{F}_i] = 0, \forall s \in \mathcal{S}, a \in \mathcal{A}, \beta \in \mathcal{B}, i \in \mathbb{N}$$

*almost surely, where $\mathcal{F}_i$ is defined in* (29)*.*

*Proof.* Let $b := (s, a, \beta)$, $\tilde{b}_i := (\tilde{s}_i, \tilde{a}_i, \beta)$ and $i \in \mathbb{N}$. We decompose the expectation using the law of total expectation to get

$$\mathbb{E}[\tilde{\phi}_i(b) | \mathcal{F}_i] = \mathbb{E}[\tilde{\phi}_i(b) | \mathcal{F}_i, \tilde{b}_i \neq b] \cdot \mathbb{P}[\tilde{b}_i \neq b | \mathcal{F}_i] + \mathbb{E}[\tilde{\phi}_i(b) | \mathcal{F}_i, \tilde{b}_i = b] \cdot \mathbb{P}[\tilde{b}_i = b | \mathcal{F}_i] \quad (30)$$

From the definition in (27), we have

$$\mathbb{E}[\tilde{\phi}_i(b) | \mathcal{F}_i, \tilde{b}_i \neq b] \cdot \mathbb{P}[\tilde{b}_i \neq b | \mathcal{F}_i] = 0$$

Then

$$
\begin{aligned}
\mathbb{E}[\tilde{\phi}_i(b) | \mathcal{F}_i, \tilde{b}_i = b] &= \mathbb{E}[(H_{\tilde{s}'}\tilde{q}_i)(b) - (H\tilde{q}_i)(b) | \mathcal{F}_i, \tilde{b}_i = b] \\
&= \xi \cdot \mathbb{E}[-(G_{\tilde{s}'}\tilde{q}_i)(b) + (G\tilde{q}_i)(b) | \mathcal{F}_i, \tilde{b}_i = b] \\
&= \xi \cdot \mathbb{E}[\mathbb{E}[-(G_{\tilde{s}'}\tilde{q}_i)(b) | \mathcal{F}_i, \tilde{b}_i = b, \tilde{\eta}_i, \mathcal{G}_{i-1}] + (G\tilde{q}_i)(b) | \mathcal{F}_i, \tilde{b}_i = b] \\
&\overset{(a)}{=} \xi \cdot \mathbb{E}[\mathbb{E}^{a,s}[-(G_{\tilde{s}_1}\tilde{q}_i)(b)] + (G\tilde{q}_i)(b) | \mathcal{F}_i, \tilde{b}_i = b] \\
&= \xi \cdot \mathbb{E}[-(G\tilde{q}_i)(b) + (G\tilde{q}_i)(b) | \mathcal{F}_i, \tilde{b}_i = b] \\
&= 0
\end{aligned}
$$

When $\tilde{s}_1$ is used in an expectation with subscript, such as $\mathbb{E}^{a,s}$, then it does not represent a sample $\tilde{s}_i$ with $i = 1$, but instead it represents the transition from $\tilde{s}_0 = s$ to $\tilde{s}_1$ distributed as $p(s, a, \cdot)$. Step (a) follows the fact the randomness of $(G_{\tilde{s}_i'}\tilde{q}_i)(b)$ only comes from $\tilde{s}_i'$ when conditioning on $\mathcal{F}_i, \tilde{b}_i = b, \tilde{\eta}_i$, and $\mathcal{G}_{i-1}$.

Then, we have

$$\mathbb{E}[\tilde{\phi}_i(b) \mid \mathcal{F}_i] = \mathbb{E}[\tilde{\phi}_i(b) \mid \mathcal{F}_i, \tilde{b}_i \neq b] \cdot \mathbb{P}[\tilde{b}_i \neq b \mid \mathcal{F}_i] + \mathbb{E}[\tilde{\phi}_i(b) \mid \mathcal{F}_i, \tilde{b}_i = b] \cdot \mathbb{P}[\tilde{b}_i = b \mid \mathcal{F}_i]$$
$$= 0 + 0 \cdot \mathbb{P}[\tilde{b}_i = b \mid \mathcal{F}_i]$$
$$= 0$$

$\square$

**Lemma B.7.** *The noise $\tilde{\phi}_i$ defined in* (27) *satisfies*

$$\mathbb{E}[(\tilde{\phi}_i(s, a, \beta))^2 | \mathcal{F}_i] \leq A + B\|\tilde{q}_i\|_\infty^2, \quad \forall s \in \mathcal{S}, a \in \mathcal{A}, \beta \in \mathcal{B}, i \in \mathbb{N}$$

*almost surely for some $A, B \in \mathbb{R}_+$, $\mathcal{F}_i$ is defined in* (29).

*Proof.* Let $b := (s, a, \beta)$, $\tilde{b}_i := (\tilde{s}_i, \tilde{a}_i, \beta)$ and $i \in \mathbb{N}$. We decompose the expectation using the law of total expectation to get

$$\mathbb{E}[\tilde{\phi}_i(b)^2 \mid \mathcal{F}_i] = \mathbb{E}[\tilde{\phi}_i(b)^2 \mid \mathcal{F}_i, \tilde{b}_i \neq b] \cdot \mathbb{P}[\tilde{b}_i \neq b \mid \mathcal{F}_i] + \mathbb{E}[\tilde{\phi}_i(b)^2 \mid \mathcal{F}_i, \tilde{b}_i = b] \cdot \mathbb{P}[\tilde{b}_i = b \mid \mathcal{F}_i]$$
$$= \mathbb{E}[\tilde{\phi}_i(b)^2 \mid \mathcal{F}_i, \tilde{b}_i = b] \cdot \mathbb{P}[\tilde{b}_i = b \mid \mathcal{F}_i]$$

This is because $\mathbb{E}[\tilde{\phi}_i(b)^2 \mid \mathcal{F}_i, \tilde{b}_i \neq b] = 0$. Let us evaluate $\mathbb{E}[\tilde{\phi}_i(b)^2 \mid \mathcal{F}_i, \tilde{b}_i = b]$.

$$\mathbb{E}[\tilde{\phi}_i(b)^2 \mid \mathcal{F}_i, \tilde{b}_i = b] = \mathbb{E}\left[\left((H_{\tilde{s}_i'}\tilde{q}_i)(b) - (H\tilde{q}_i)(b)\right)^2 \mid \mathcal{F}_i, \tilde{b}_i = b\right]$$
$$= \mathbb{E}\left[\left(-\xi \cdot (G_{\tilde{s}_i'}\tilde{q}_i)(b) + \xi \cdot (G\tilde{q}_i)(b)\right)^2 \mid \mathcal{F}_i, \tilde{b}_i = b\right]$$
$$= \xi^2 \cdot \mathbb{E}\left[\mathbb{E}\left[\left(-(G_{\tilde{s}_i'}\tilde{q}_i)(b) + (G\tilde{q}_i)(b)\right)^2 \mid \mathcal{F}_i, \tilde{b}_i = b, \mathcal{G}_{i-1}\right] \mid \mathcal{F}_i, \tilde{b}_i = b\right]$$

Let us define $\tilde{\delta}_i(s', \beta)$ in (31).

$$\tilde{\delta}_i(s', \beta) = r(s, a, s') + \max_{a' \in \mathcal{A}} \tilde{q}_i(s', a', \beta) - \tilde{q}_i(s, a, \beta) \tag{31}$$

$$\xi^2 \cdot \mathbb{E}\left[\mathbb{E}\left[\left(-(G_{\tilde{s}_i'}\tilde{q}_i)(b) + (G\tilde{q}_i)(b)\right)^2 \mid \mathcal{F}_i, \tilde{b}_i = b, \tilde{\eta}_i, \mathcal{G}_{i-1}\right] \mid \mathcal{F}_i, \tilde{b}_i = b\right]$$
$$\overset{(a)}{=} \xi^2 \cdot \mathbb{E}\left[\mathbb{E}^{a,s}\left[\left((G_{\tilde{s}_1}\tilde{q}_i)(b) - (G\tilde{q}_i)(b)\right)^2\right] \mid \mathcal{F}_i, \tilde{b}_i = b\right]$$
$$\overset{(b)}{=} \xi^2 \cdot \mathbb{E}\left[\mathbb{E}^{a,s}\left[\left(\mathbb{E}[\partial\ell_\beta(\tilde{\delta}_i(\tilde{s}_1, \beta)) \mid \tilde{s}_1] - \mathbb{E}^{a,s}[\partial\ell_\beta(\tilde{\delta}_i(\tilde{s}_1, \beta))]\right)^2\right] \mid \mathcal{F}_i, \tilde{b}_i = b\right]$$
$$\overset{(c)}{=} \xi^2 \cdot \mathbb{E}\left[\mathbb{E}^{a,s}\left[\left(\mathbb{E}[\partial\ell_\beta(\tilde{\delta}_i(\tilde{s}_1, \beta)) \mid \tilde{s}_1]\right)^2\right] - \left(\mathbb{E}^{a,s}[\partial\ell_\beta(\tilde{\delta}_i(\tilde{s}_1, \beta))]\right)^2\right] \mid \mathcal{F}_i, \tilde{b}_i = b\right]$$
$$\leq \xi^2 \cdot \mathbb{E}\left[\mathbb{E}^{a,s}\left[\left(\mathbb{E}[\partial\ell_\beta(\tilde{\delta}_i(\tilde{s}_1, \beta)) \mid \tilde{s}_1]\right)^2\right] \mid \mathcal{F}_i, \tilde{b}_i = b\right]$$
$$\overset{(d)}{\leq} \xi^2 \cdot \mathbb{E}\left[\max_{s' \in \mathcal{S}} \partial\ell_\beta(\tilde{\delta}_i(s', \beta))^2 \mid \mathcal{F}_i, \tilde{b}_i = b\right]$$

$$\overset{(e)}{\leq} \xi^2 \cdot \mathbb{E}\left[\max_{s' \in \mathcal{S}}(|\ \partial \ell_\beta(\tilde{\delta}_i(s', \beta)) - \partial \ell_\beta(0)\ |)^2 \mid \mathcal{F}_i, \tilde{b}_i = b\right]$$

$$\overset{(f)}{\leq} \xi^2 \cdot \mathbb{E}\left[\max_{s' \in \mathcal{S}}(\frac{\beta}{e^{\beta z_{\min}}} \cdot |\ \tilde{\delta}_i(s', \beta)\ |)^2 \mid \mathcal{F}_i, \tilde{b}_i = b\right]$$

$$\leq \xi^2 \cdot (\frac{\beta}{e^{\beta z_{\min}}})^2 \cdot \mathbb{E}\left[\max_{s' \in \mathcal{S}}(\tilde{\delta}_i(s', \beta))^2 \mid \mathcal{F}_i, \tilde{b}_i = b\right]$$

$$\overset{(g)}{\leq} \xi^2 \cdot (\frac{\beta}{e^{\beta z_{\min}}})^2 \cdot (2 \cdot \|r\|_\infty^2 + 8 \cdot \|\tilde{q}_i\|_\infty^2)$$

Step $(a)$ follows Lemma B.5 given that the randomness of $-(G_{\tilde{s}_i'}\tilde{q}_i)(b) + (G\tilde{q}_i)(b)$ only comes from $\tilde{s}_i'$ when conditioning on $\mathcal{F}_i, \tilde{b}_i = b, \tilde{\eta}_i$ and $\mathcal{G}_{i-1}$. Step $(b)$ follows by substituting $(G_{\tilde{s}_i'}\tilde{q}_i)(b)$ with (12), substituting $(G\tilde{q}_i)(b)$ with (13), and replacing by using $\tilde{\delta}_i$ defined in (31). The equality in step $(c)$ holds because for a random variable $\tilde{x} = \mathbb{E}[\partial \ell_\beta(\tilde{\delta}_i(\tilde{s}_1, \beta)) \mid \tilde{s}_1]$, the variance satisfies $\mathbb{E}[(\tilde{x} - \mathbb{E}[\tilde{x}])^2] = \mathbb{E}[\tilde{x}^2] - (\mathbb{E}[\tilde{x}])^2$. Step $(d)$ upper bounds the expectation by a maximum. Step $(e)$ uses $\partial \ell_\beta(0) = 0$ from the definition in (9). Step $(f)$ uses Lemma 4.4 to bound the derivative difference as a function of the step size. Step $(g)$ derives the final upper bound since

$$\|r\|_\infty = \max_{s, s' \in \mathcal{S}, a \in \mathcal{A}}|r(s, a, s')|, \quad \|\tilde{q}_i\|_\infty = \max_{s \in \mathcal{S}, a \in \mathcal{A}}|\tilde{q}_i(s, a, \beta)|$$

$$
\begin{aligned}
\max_{s' \in \mathcal{S}} \tilde{\delta}_i(s', \beta)^2 &\leq (\|r\|_\infty + 2\|\tilde{q}_i\|_\infty)^2 \\
&\leq (\|r\|_\infty + 2\|\tilde{q}_i\|_\infty)^2 + (\|r\|_\infty - 2\|\tilde{q}_i\|_\infty)^2 \\
&= 2\|r\|_\infty^2 + 8\|\tilde{q}_i\|_\infty^2
\end{aligned}
\tag{32}
$$

Then, we have

$$
\begin{aligned}
\mathbb{E}[\tilde{\phi}_i(b)^2 \mid \mathcal{F}_i] &= \mathbb{E}[\tilde{\phi}_i(b)^2 \mid \mathcal{F}_i, \tilde{b}_i = b] \cdot \mathbb{P}[\tilde{b}_i = b \mid \mathcal{F}_i] \\
&\leq \xi^2 \cdot (\frac{\beta}{e^{\beta z_{\min}}})^2 \cdot (2 \cdot \|r\|_\infty^2 + 8 \cdot \|\tilde{q}_i\|_\infty^2)
\end{aligned}
$$

Therefore, $A = \xi^2 \cdot (\frac{\beta}{e^{\beta z_{\min}}})^2 \cdot (2 \cdot \|r\|_\infty^2)$ and $B = 8\xi^2 \cdot (\frac{\beta}{e^{\beta z_{\min}}})^2$. $\qquad \square$

### B.2.3 Monotonicity of Operator $H$

Let us prove that the operator $H$ defined in (14) satisfies monotonicity.

*Proof of Lemma 3.4.* First, we prove part $(a)$: fix $\beta > 0$ and some $b = (s, a, \beta), s \in \mathcal{S}, a \in \mathcal{A}$. Fix $q$ and define

$$f(y) = \mathbb{E}^{s,a}\left[\ell_\beta(r(s, a, \tilde{s}_1) + \max_{a' \in \mathcal{A}} q(\tilde{s}_1, a', \beta) - y)\right] \tag{33}$$

The function $f$ is strongly convex with a Lipschitz-continuous gradient with parameters $\ell$ and $L$ based on Lemma 4.4. Let $y^\star = \arg\min_{y \in \mathbb{R}} f(y)$ and $\exists l \in [1/L, 1/\ell]$ such that

$$
\begin{aligned}
(Hq)(b) &= (q - \xi Gq)(b) \\
&\overset{(a)}{=} q(b) - \xi f'(q(b)) \\
&\overset{(b)}{=} (1 - \xi/l)q(b) + \xi/l \cdot y^\star \\
&\overset{(c)}{=} (1 - \xi/l)q(b) + \xi/l \cdot (\hat{B}q)(b)
\end{aligned}
\tag{34}
$$

Step (a) follows the replacement with (12) and (33). Step(b) follows the Lemma 2.1. Step(c) follows the fact that $\hat{B}$ is the ERM Bellman operator defined in (11) and $y^\star$ is the unique solution to (10).

Given $x(b) \leq y(b)$, let us prove that $(Hx)(b) \leq (Hy)(b)$. Since the ERM Bellman operator $\hat{B}$ is monotone, we have

$$x(b) - y(b) \leq 0 \Rightarrow (\hat{B}x)(b) - (\hat{B}y)(b) \leq 0$$

Then we have

$$(Hx)(b) - (Hy)(b) = (1 - \xi/l)x(b) + \xi/l \cdot (\hat{B}x)(b) - \big((1 - \xi/l)y(b) + \xi/l \cdot (\hat{B}y)(b)\big)$$
$$= (1 - \xi/l)(x(b) - y(b)) + \xi/l \cdot ((\hat{B}x)(b) - (\hat{B}y)(b))$$
$$\leq 0$$

Second, let us prove part $(b)$: $(Hq)(b)$ can be written as follows.

$$(Hq)(b) = (1 - \xi/l)q(b) + \xi/l \cdot (\hat{B}q)(b)$$

From [44, theorem 3.3], we know that $q^\star(b)$ is a fixed point of $\hat{B}$. Then $q^\star(b)$ is also a fixed point of $H$. That is, $(Hq^\star)(b) = q^\star(b)$.

Third, let us prove part $(c)$, we omit $b$ in the $(Hq)(b)$ and rewrite it as follows.

$$Hq = (1 - \xi/l)q + \xi/l \cdot \hat{B}q$$

Given $e \in \mathbb{R}^n$ is the vector with all components equal to 1 and if $c$ is a positive scalar, we show that

$$Hq - c \cdot e = H(q - c \cdot e) \leq H(q + c \cdot e) = Hq + c \cdot e$$

1) Let us prove $Hq - c \cdot e = H(q - c \cdot e)$

$$Hq - c \cdot e = (1 - \xi/l) \cdot q + (\xi/l) \cdot \hat{B}q - c \cdot e$$
$$= (1 - \xi/l) \cdot q - (1 - \xi/l) \cdot c \cdot e + (\xi/l) \cdot \hat{B}q - (\xi/l) \cdot c \cdot e$$
$$= (1 - \xi/l)(q - c \cdot e) + (\xi/l)(\hat{B}q - c \cdot e)$$
$$\stackrel{(a)}{=} (1 - \xi/l)(q - c \cdot e) + (\xi/l)(\hat{B}(q - c \cdot e))$$
$$= H(q - c \cdot e)$$

Step $(a)$ follows from the law invariance property of ERM [19].

2) Let us prove $H(q - c \cdot e) \leq H(q + c \cdot e)$

Because $q - c \cdot e \leq q + c \cdot e$ and the part $(a)$ of the operator $H$, we have

$$H(q - c \cdot e) \leq H(q + c \cdot e)$$

3) Let us prove $H(q + c \cdot e) = H(q) + c \cdot e$

$$H(q + c \cdot e)$$
$$= (1 - \xi/l)(q + c \cdot e) + (\xi/l)\hat{B}(q + c \cdot e)$$
$$= (1 - \xi/l)(q + c \cdot e) + (\xi/l)(\hat{B}(q) + c \cdot e)$$
$$\stackrel{(a)}{=} (1 - \xi/l)(q + c \cdot e) + (\xi/l)\hat{B}(q) + (\xi/l) \cdot c \cdot e$$
$$= (1 - \xi/l)q + (1 - \xi/l) \cdot (c \cdot e) + (\xi/l) \cdot c \cdot e + (\xi/l)\hat{B}(q)$$
$$= (1 - \xi/l)q + c \cdot e + (\xi/l)\hat{B}(q)$$
$$= H(q) + c \cdot e$$

Step $(a)$ follows from the law invariance of ERM. Then we have

$$Hq - c \cdot e \leq H(q - c \cdot e) \leq H(q + c \cdot e) \leq Hq + c \cdot e$$

$\square$

### B.2.4   Proof of Theorem 4.2

*Proof.* Proof of Theorem 4.2  We verify that the sequence of our Q-learning iterates satisfies the properties in Proposition B.3. The step size condition in Theorem 4.2 guarantees that we satisfy property $(a)$ in Proposition B.3. Lemma B.6 and Lemma B.7 show that we satisfy property (b) in Proposition B.3. Lemma 3.4 shows that we satisfy property $(c)$ in Proposition B.3.  $\square$

### B.3 Proof of Lemma 4.3

*Proof.* First, from Theorem 3.3, we have

$$B_\beta q = \hat{B}_\beta q$$
$$\Downarrow$$
$$\text{ERM}_\beta^{a,s}\left[r(s,a,\tilde{s}_1) + \max_{a' \in \mathcal{A}} q(\tilde{s}_1, a', \beta)\right] = \arg\min_{y \in \mathbb{R}} \mathbb{E}^{a,s}\left[\ell_\beta\left(r(s,a,\tilde{s}_1) + \max_{a' \in \mathcal{A}} q(\tilde{s}_1, a', \beta) - y\right)\right]$$

For the part $(a)$, ERM is monotone [19]. That is, for some fixed $\beta \in \mathcal{B}$ value, if $x \leq y$, then $\hat{B}_\beta x \leq \hat{B}_\beta y$.

For the part $(b)$, $\ell_\beta$ is a strongly convex function because $\ell_\beta''(y) = \beta \cdot e^{-\beta \cdot (\tilde{x} - y)} > 0$. Then there will be a unique $y^\star$ value such that $\mathbb{E}^{a,s}\left[\ell_\beta\left(r(s,a,\tilde{s}_1) + \max_{a' \in \mathcal{A}} q(\tilde{s}_1, a', \beta) - y\right)\right]$ attains the minimum value, and the $y^\star$ value is equal to $q^\star(s,a,\beta)$, which follows from Proposition 3.2. So $\hat{B}_\beta q^\star = q^\star$

For the part $(c)$, ERM is monotone and satisfies the law-invariance property [19].

$$\hat{B}_\beta(q) - 1 \cdot g \overset{(1)}{\leq} \hat{B}_\beta(q - 1 \cdot g) \overset{(2)}{\leq} \hat{B}_\beta(q + 1 \cdot g) \overset{(3)}{\leq} \hat{B}_\beta(q) + 1 \cdot g$$

Steps (1) and (3) follow from the law-invariance property of ERM. Step (2) follows from the mononotone property of ERM. $\square$

### B.4 Proof of Lemma 4.4

*Proof.* We use the fact that $\ell_\beta$ is twice continuously differentiable and prove the $l$-strong convexity from [33, theorem 2.1.11]:

$$l = \inf_{z \in [z_{\min}, z_{\max}]} \ell_\beta''(z) = \inf_{z \in [z_{\min}, z_{\max}]} \beta \exp\left(-\beta z\right) = \beta \exp\left(-\beta z_{\max}\right).$$

To prove $L$-Lipschitz continuity of the derivative, using [37, Theorem 9.7], we have that

$$L = \sup_{z \in [z_{\min}, z_{\max}]} |\ell_\beta''(z)| = \sup_{z \in [z_{\min}, z_{\max}]} |\beta \exp\left(-\beta z\right)| = \beta \exp\left(-\beta z_{\min}\right).$$

$\square$

### B.5 Proof of Corollary 4.5

*Proof.* From Theorem 4.2, for some $\beta \in \mathcal{B}$, ERM-Q learning algorithm converges to the optimal ERM value function. Theorem 3.5 shows that there exists $\delta-$optimal policy such that it is an ERM-TRC optimal for some $\beta \in \mathcal{B}$. It is sufficient to compute an ERM-TRC optimal policy for one of those $\beta$ values. This analysis shows that Algorithm 2 converges to its optimal EVaR value funciton. $\square$

## C Additional Material of Section 5

The machine used to conduct all experiments referenced in Section 5 is a single machine with the following specifications:

- AMD Ryzen Thread ripper 3970X 32-Core (64) @ 4.55 GHz
- 256 GB RAM
- Julia 1.11.5

