# OpenReview forum: "Risk-Averse Total-Reward Reinforcement Learning"
_NeurIPS.cc/2025/Conference — NeurIPS 2025 poster_

### Official Review · Reviewer_Cf62 · 2025-06-27

**Clarity:** 3
**Significance:** 2
**Originality:** 3
**Rating:** 5
**Confidence:** 3

**Summary:**

This paper proposes Q-learning algorithms for optimizing risk-averse objectives (specifically, the entropic risk measure (ERM) and entropic value-at-risk (EVaR)) under the total reward criterion in transient, undiscounted MDPs. The authors derive convergence guarantees using the elicitability property of ERM and present numerical experiments on tabular domains.

**Questions:**

- Please provide clarification concerning the impact of truncation in the Cliffwalking experiments (concern 1).

Additionally, I have a few clarification questions to clear up confusions:
- Sec1, line 45: the authors mention that "Generalizing the standard Q-learning algorithm to risk-averse objectives requires computing the full return distribution rather than just its expectation". Could you use the distributional Bellman operator using e.g., C51 [1], learn the full distribution over returns, and use this distribution to learn the optimal risk averse policy for a specific $\alpha$?
- Out of curiosity, since the MDP is transient, and that any policy eventually reaches the sink state (where there is zero reward), isn't this equivalent to the episodic setting?

**Ethical Concerns:**

["NO or VERY MINOR ethics concerns only"]

**Final Justification:**

- The authors present a rigorous and well-motivated extension of Q-learning to risk-sensitive objectives in an undiscounted, infinite-horizon setting, with convergence guarantees.
- The algorithm and its theoretical background are generally well explained, and the exposition is mostly clear.

During the rebuttal:
- The authors addressed the difficulty of using the distributional Bellman equation
- Explained the reason for the cutoff at 20k steps for the cliffwalking experiments

My reasons for not scoring higher include:
- heterogeneous experimental setup, making it hard to make generalizations or comparisons across experiments
- the cutoff in cliffwalking experiments resulting in not all episodes ending in the sink-state. This has been partially addressed during the rebuttal, the authors mentioning that with the current cutoff the difference between the EVaR of the simulated returns with their true model-based EVaR is minimal

**Limitations:**

Yes

**Quality:**

3

**Strengths And Weaknesses:**

Strengths
---------
- The authors present a rigorous and well-motivated extension of Q-learning to risk-sensitive objectives in an undiscounted, infinite-horizon setting, with convergence guarantees.
- The algorithm and its theoretical background are generally well explained, and the exposition is mostly clear.

Concerns
---------
- In the cliff-walk experiment, the sink state should always be reached due to random chance (the probability of 0.09 of taking a different action than selected), where the agent receives a reward of 2. However, the evaluation is bounded at 20k steps. For $\alpha=0.6$, did the agent actually learn the optimal policy? Could it be that the mean reward of $−0.074$ is solely due to episode truncation? If so, the experiment does not properly reflect the tackled setting (undiscounted infinite-horizon). Then, it would be valuable to change the setting such that episode truncation does not come into play, e.g., by increasing the probability of taking a different action than selected.
- The choice of $\alpha$ in different experiments (e.g., $\alpha=0.3$ for stability plots, vs. $\alpha=0.2,0.6$ elsewhere) seems arbitrary. Since the authors already run simulations for $\alpha=0.2,0.6$, it would be more consistent and informative to reuse them in all plots.

---

> ### Author Rebuttal · Authors · 2025-07-30
>
> Rebuttal
>
> We sincerely appreciate the detailed review and thoughtful comments.
>
> ***In the cliff-walk experiment, the sink state should always be reached due to random chance (the probability of 0.09 of taking a different action than selected), where the agent receives a reward of 2. However, the evaluation is bounded at 20k steps. For $\alpha = 0.6$, did the agent actually learn the optimal policy? Could it be that the mean reward of is solely due to episode truncation? If so, the experiment does not properly reflect the tackled setting (undiscounted infinite-horizon). Then, it would be valuable to change the setting such that episode truncation does not come into play, e.g., by increasing the probability of taking a different action than selected.Please provide clarification concerning the impact of truncation in the Cliffwalking experiments (concern 1).***
>
> This is a great suggestion! For $\alpha = 0.6$, the agent learns the optimal policy for 20k steps. If we keep increasing the number of steps, the agent will eventually enter the sink state and receive the reward of 2, and the mean reward will increase. The mean reward is affected by both the episode truncation and the optimal policy.
>
> We will increase the probability of taking a different action and include a comparison of return distributions in the final version.
>
>
> ***The choice of $\alpha$ in different experiments (e.g., $\alpha = 0.3$ for stability plots, vs. $\alpha = 0.2,0.6 elsewhere) seems arbitrary. Since the authors already run simulations for , it would be more consistent and informative to reuse them in all plots.***
>
> Thanks! We will choose consistent $\alpha$ values for all plots in the final version.
>
> ***Sec1, line 45: the authors mention that "Generalizing the standard Q-learning algorithm to risk-averse objectives requires computing the full return distribution rather than just its expectation". Could you use the distributional Bellman operator using e.g., C51 [1], learn the full distribution over returns, and use this distribution to learn the optimal risk averse policy for a specific $\alpha$?***
>
> It is a great question! The standard Bellman operator differs fundamentally from the distributional Bellman operator, which has three sources of randomness. They are the randomness in the reward, the randomness in the transition, and the next-state value distribution. We define the standard ERM Bellman operator and leverage the elicitability property of ERM to define a new ERM Bellman operator as the minimizer of the exponential loss function, and prove the equivalence of the two Bellman operators and the existence of a fixed point. We utilize the new ERM Bellman operator to develop the proposed Q-learning algorithms and compute the optimal risk-averse policy for a specific value of $\alpha$.
>
> Recent work (Shiau Hong Lim,Ilyas Malik. Distributional Reinforcement Learning for Risk-Sensitive Policies. NeurIPS 2022) addresses the problem of learning a risk-sensitive policy based on the CVaR risk measure using distributional reinforcement learning. In particular, they show that the standard action-selection strategy when applying the distributional Bellman optimality operator can result in convergence to neither the dynamic, Markovian CVaR nor the static, non-Markovian CVaR. They introduce a new distributional Bellman operator, but the convergence properties of the proposed distributional Bellman operator remain unknown in the general setting.
>
> Our paper focuses on the ERM-TRC and EVaR-TRC objectives. It is unclear how to modify the regular distributional Bellman operator to fit with ERM and EVaR risk measures. The convergence properties of the modified distributional Bellman operator may be unknown in the general setting. This can be an interesting problem for future work.
>
> ***Out of curiosity, since the MDP is transient, and that any policy eventually reaches the sink state (where there is zero reward), isn't this equivalent to the episodic setting?***
>
> Good question! Our setting is very similar to episodic RL. We use the term TRC because it is more specific and precise. We will clarify this point in the paper.
>
> In our understanding of the episodic setting, the agent interacts with the environment for a fixed number of steps, and the episode ends after a certain number of steps or when a terminal state is reached. Each episode has a clear start and end. Episodes are usually finite. When an episode is forcibly halted, the last state is not considered a sink state.
>
> For TRC criterion,  the start state is uniformly distributed among the non-sink states, as stated in assumption 2.1 of the reference [41]. Also, the number of steps to reach the goal state is not fixed and can vary significantly. The probability of eventually reaching the goal state is 1, but the expected number of steps required to do so is not bounded.

---

> > ### Comment · Reviewer_Cf62 · 2025-08-04
> >
> > Thank you for your answer, they clarify some of the doubts I had. They also raised a few follow-up questions:
> >
> > > For $\alpha=0.6$, the agent learns the optimal policy for 20k steps.
> >
> > So, from what I understand, the learned policy is different from the optimal policy in the infinite-horizon setting. One of the reasons for numerical evaluations is to empirically confirm the theoretical contributions. Since this work tackles infinite-horizon, undiscounted settings, I believe the experiments should reflect this. You mention that "If we keep increasing the number of steps, the agent will eventually enter the sink state and receive the reward of 2, and the mean reward will increase". But this has not been actually confirmed, and so we do not know that the proposed algorithm will in fact reach the optimal policy under the setting it has been designed for. This greatly reduces the relevance of the numerical evaluations.
> >
> > > The standard Bellman operator differs fundamentally from the distributional Bellman operator, which has three sources of randomness. They are the randomness in the reward, the randomness in the transition, and the next-state value distribution.
> >
> > I am not sure how to understand this statement. Randomness in reward an transition function is inherent to any MDP, they are soures of randomness for the standard Bellman operator as well. Could you clarify what you mean here, and their consequences for learning optimal averse policies under a specific $\alpha$?

---

> > > ### Author Response · Authors · 2025-08-04
> > >
> > > **Re: Then, it would be valuable to change the setting such that episode truncation does not come into play.**
> > >
> > > Thanks for following up. We feel that our prior response was unclear, which may have caused some unintended confusion. Our apologies.  Let us try to clarify.
> > >
> > > First, we concur that the concern over the impact of policy truncation is significant in the TRC objective; much more so than in discounted objectives.
> > >
> > > Second, the primary purpose of simulating the policies was to obtain the return distributions depicted in Figure 3 rather than to confirm convergence.
> > >
> > > Third, the primary evidence of the convergence to an optimal policy is the convergence of the q-functions to the optimal model-based q-functions shown in Figures 5 and 6. The simulation results represent only secondary evidence, primarily serving as a sanity check to confirm the results using a technique that does not rely on the risk-averse Bellman equations.
> > >
> > > To understand whether truncating the horizon impacts the estimated return, we compared the EVaR of the simulated returns with their true model-based EVaR (which we computed by solving a series of systems of linear equations in Eq. (6)). The difference between the two EVaR estimates was negligible for episodes that were truncated after 20,000 steps, which is why we chose this cutoff.
> > >
> > > Thanks for raising this valid concern. To address it, we plan on also reporting the difference between the simulated EVaR estimates and model-based EVaR q-functions. We will also make it clear that the primary purpose of the simulations is to estimate the return distribution rather than to confirm convergence.
> > >
> > > **Could you use the distributional Bellman operator using e.g., C51 [1], learn the full distribution over returns, and use this distribution to learn the optimal risk-averse policy?**
> > >
> > > Repurposing the distributional Bellman operator for optimizing EVaR-TRC appears to be very difficult, if not impossible.
> > >
> > > The first fundamental complication is that in the TRC (as opposed to discounted criteria), the return random variable has infinite support. There may be a positive probability of attaining an arbitrarily large or an arbitrarily small cumulative reward, which may ultimately cancel out. Although one could truncate the support of this random variable, it is difficult to do so without introducing an arbitrarily large error.
> > >
> > > The second fundamental complication arises when making decisions with return distributions. The distributional Bellman operator must choose an action by comparing the distribution of their return. How to choose such an action is a critical and non-trivial design decision that previous efforts often fail to address properly. While it is appealing to try to repurpose distributional Bellman operators for risk aversion, we are only aware of earlier efforts that suffer from fundamental flaws. For more discussion of these issues, please see [Hong & Malik, 2022] for CVaR and [Hau et al., Q-learning for Quantile MDPs: A Decomposition, Performance, and Convergence Analysis, 2025] for VaR. Using distributional RL to optimize EVaR would fail in the same way as the efforts for VaR and CVaR previously because EVaR is not dynamically consistent.
> > >
> > > This is an important point, and we will add a brief discussion to the paper outlining the challenges with adapting distributional RL to the risk-averse TRC setting.

---

> > > > ### Comment · Reviewer_Cf62 · 2025-08-08
> > > >
> > > > Thank you for the clarifications. The choice for the cutoff is much clearer now, even though I still believe that it would be interesting to have the return distribution for the theoretical setting (ie, each episode ends in the sink state).
> > > >
> > > > Your comments for the distributional Bellman equation are also interesting, thank you for sharing them.
> > > >
> > > > Overall, I find this a technically solid paper, that I recommend for acceptance.

---

### Official Review · Reviewer_xc6u · 2025-06-28

**Clarity:** 4
**Significance:** 4
**Originality:** 3
**Rating:** 4
**Confidence:** 4

**Summary:**

This paper introduces ERM-TRC Q-learning which is a model-free RL algorithm in the risk-averse total reward setting using the ERM and EVaR objectives. Finding an optimal policy in the absence of knowledge of transition probabilities (which makes estimating risk averse rewards hard) is the key problem addressed in the paper. The paper proposes a novel Bellman operator leveraging the elicitability property of ERM, enabling a stochastic gradient descent approach for Q-learning. The work guarantees asymptotic almost sure convergence to the optimal policy. Technical challenges in algorithm design and establishing convergence guarantees include (a) generalizing the standard Q-learning algorithm to risk-averse objectives requiring computing the full return distribution rather than just its expectation; (b) risk-averse TRC Bellman operator not being a contraction (c) boundedness of the risk-averse objectives.

**Questions:**

N/A

**Ethical Concerns:**

["NO or VERY MINOR ethics concerns only"]

**Final Justification:**

Why Weak Accept (4): Important problem, application of elicitability for establishing guarantees of Q-learning given the challenges of non-contraction through the elegant Bellman operator $\hat{B}_{\beta}$

Why not Accept (5): no comparison against model-based algorithm baselines in the experiments which also raises concerns about the practical applicability of the algorithm

**Limitations:**

Yes, the authors have adequately addressed the limitations and potential negative societal impact of their work.

**Quality:**

3

**Strengths And Weaknesses:**

**Strengths:**
- Risk-Averse Total Reward Reinforcement Learning is an important problem to consider and the total reward criterion poses significant theoretical challenges in algorithm development and analysis
- While elicitability has been explored for risk measures, its application to derive Q-learning in the undiscounted total-reward setting is novel given the challenges of non-contraction
- Elegant definition of the Bellman operator $\hat{B}_{\beta}$ amenable to stochastic gradient descent
- Well presented, intuitive structure of the algorithms and guarantees


**Weaknesses:**
- The paper considers only the tabular setting - extending the work to function approximation based methods and model-free gradient based methods will make it stronger
- While, I understand that this is primarily a theoretical work, the experiments are rather limited lacking evaluation on tasks with larger state-action spaces, function approximation and no comparison against existing model-based algorithm baselines
- As the authors acknowledge in the conclusion, the assumption that $\beta$ has be to small enough and estimation of $z_{min}, z_{max}$ limits the practical applicability of the algorithm

---

> ### Author Rebuttal · Authors · 2025-07-30
>
> Rebuttal:
>
> We sincerely appreciate the detailed review and thoughtful comments.
>
>
> ***The paper considers only the tabular setting - extending the work to function approximation based methods and model-free gradient based methods will make it stronger***
>
> Thanks for making this point. We would like to clarify that the proposed algorithm is not limited to tabular representations. The algorithm is formulated for arbitrary model-free, gradient-based methods and can be used with any differentiable value function approximator. The tabular implementation serves as a concrete instantiation for experimental validation and aims to provide a clear and controlled validation that the value function learned by Algorithms 1 and 2 converges to the true optimal value function, consistent with the theoretical guarantees (see reference [41] in the paper).
>
> We agree that extending the work to value function approximation and deep reinforcement learning methods is a valuable direction and would further demonstrate the scalability and generality of our approach. We intend to explore these extensions in future work.
>
>
> ***As the authors acknowledge in the conclusion, the assumption that $\beta$  has be to small enough and estimation of $z_{min}, z_{max}$ limits the practical applicability of the algorithm***
>
> We expect to leverage existing algorithms of refining temporal difference(TD) bounds to improve practical efficacy of the proposed algorithms. We will discuss possible heuristics to develop a better methodology for establishing these parameters in the future work section of the paper.

---

> ### Comment · Reviewer_xc6u · 2025-08-03
>
> Thank you for the response!

---

### Official Review · Reviewer_TEZu · 2025-07-03

**Clarity:** 2
**Significance:** 3
**Originality:** 3
**Rating:** 5
**Confidence:** 3

**Summary:**

The paper proposes a risk-averse total-reward MDP framework and a Q-learning algorithm for learning optimal policies under both ERM and EVaR objectives. It provides theoretical analysis on the uniqueness of the solution, the convergence of the algorithm, etc. The effectiveness of the approach is demonstrated through two tabular task examples. This paper seems a direct extension of [1] (reference [41] in the paper).

[1] Su, X., Petrik, M., & Grand-Clément, J. (2025). Risk-averse Total-reward MDPs with ERM and EVaR. Proceedings of the AAAI Conference on Artificial Intelligence, 39(19), 20646-20654.

**Questions:**

* In Section 5, the authors compare two optimal policies derived under different risk preference levels (\alpha). The results show that with \alpha = 0.6, the returns have a wider range and a lower mean, while the policy with \alpha=0.2 achieves better overall performance with a similar standard deviation. In general, a controller tradeoffs expected total rewards against risk, and reducing risk often comes at the cost of performance. However, in this case, the more risk-averse policy appears to perform better with even lower risk. I suspect this may be due to the setting where the agent does not incur a penalty for remaining in the grid. Could the authors consider adding a small penalty for staying in place and then show how the expected total rewards and EVaR change as a function of \alpha?

* In practical applications, what guidance can the authors provide on selecting or tuning the hyperparameters for these optimization problems? Are there general strategies or heuristics that work well in practice?

**Ethical Concerns:**

["NO or VERY MINOR ethics concerns only"]

**Final Justification:**

I decided to keep my rating. The authors answered my questions in their response regarding the risk preference parameter. I have no more questions.

**Limitations:**

Agree with the authors on Limitations in Section 6.

**Quality:**

4

**Strengths And Weaknesses:**

Strengths:

* The paper is clearly written and has a clear logical structure.
* It offers a comprehensive theoretical analysis of the proposed algorithms, covering the problem formulation, uniqueness of the solution, convexity properties, the relationship between the EVaR-TRC and ERM-TRC problems, and convergence guarantees.

Weaknesses:

* Some technical details are missing, which may make it difficult for readers unfamiliar with prior work to fully understand the paper. While space limitations are understandable, including a more detailed sketch of key proofs would help improve clarity and accessibility.

---

> ### Author Rebuttal · Authors · 2025-07-30
>
> Rebuttal:
>
> We sincerely appreciate the detailed review and thoughtful comments.
>
> ***Some technical details are missing, which may make it difficult for readers unfamiliar with prior work to fully understand the paper. While space limitations are understandable, including a more detailed sketch of key proofs would help improve clarity and accessibility.***
>
> Thanks! We will add a more detailed sketch of key proofs in the main text in the final version of the paper. We will also restate the key theorems we use from the work [41] and provide a detailed explanation of the key connections between [41] and our work in the appendix of the final version.
>
>
> ***In Section 5, the authors compare two optimal policies derived under different risk preference levels (\alpha). The results show that with \alpha = 0.6, the returns have a wider range and a lower mean, while the policy with \alpha=0.2 achieves better overall performance with a similar standard deviation. In general, a controller trades off expected total rewards against risk, and reducing risk often comes at the cost of performance. However, in this case, the more risk-averse policy appears to perform better with even lower risk. I suspect this may be due to the setting where the agent does not incur a penalty for remaining in the grid. Could the authors consider adding a small penalty for staying in place and then show how the expected total rewards and EVaR change as a function of $\alpha$?***
>
>
> This is a great suggestion, thank you. First, let us review the numerical results from the closest related work(reference [41] in the paper). When the risk level $\alpha$ increases from 0.2 to 0.7, the probability of getting the highest total reward increases. That is, reducing risk comes at the cost of performance. However, when the risk level $\alpha=0.9$, the probability of getting the highest total reward is lower than the probability of getting the highest total reward with $\alpha=0.7$. It is unclear what the risk level $\alpha$ would be to maximize the probability of achieving the highest total reward in the work [41].
>
> Second, the q-value functions computed by the proposed Q-learning algorithms converge to the optimal value functions computed by the linear programming in reference [41]. Our numerical results exhibit similar behaviors in the total reward criterion setting.
>
> Third, we will add a small penalty for staying in place and include the results of the expected total rewards and EVaR change as a function of $\alpha$  in the final version.
>
> ***In practical applications, what guidance can the authors provide on selecting or tuning the hyperparameters for these optimization problems? Are there general strategies or heuristics that work well in practice?***
>
> This is a great question. In our numerical results, a decaying learning rate would be more effective than a fixed learning rate. Regarding the convergence results in Section 5, the learning rate with harmonic decay functions outperforms that with exponential decay functions. Also, we saw that the choice of the bounds $z_{min}$ and $z_{max}$ did not affect the convergence rate of our Q-learning algorithm significantly. We believe that more numerical experiments will be needed to understand the practical efficacy of specific strategies. Still, we expect that the general guidance for standard temporal difference bounds is likely to apply in our setting.

---

> > ### Comment · Reviewer_TEZu · 2025-08-04
> >
> > I would like to thank the authors for their detailed response. I decided to keep my rating.

---

### Official Review · Reviewer_27u3 · 2025-07-03

**Clarity:** 2
**Significance:** 3
**Originality:** 3
**Rating:** 4
**Confidence:** 3

**Summary:**

The authors proposed a model-free Q-learning algorithm for the Risk-Averse problem with ERM and EVAR metrics. To address the issue of unknown transition, the authors employed the elicitability property to replace the Bellman update with a gradient descent update, thereby proving the convergence of the proposed algorithm for both risk metrics.

**Questions:**

Please refer to the Strengths and weaknesses.

**Ethical Concerns:**

["NO or VERY MINOR ethics concerns only"]

**Final Justification:**

The authors clearly stated the novelty and contributions of the paper. Thus, I raise my score of originality to 3 and final rating to 4.

**Limitations:**

yes

**Quality:**

2

**Strengths And Weaknesses:**

Quality:2

The main idea of this paper is that the risk-averse MDP with total reward satisfies a Bellman equation with the modified Bellman operator. However, unlike in the discounted case, the Bellman operator is not a contraction. Thus, the authors utilized the monotonicity property and followed the proof idea from standard stochastic iterative algorithms.

The proof seems solid to me. However, I have a question about the assumption. In the paper, the authors mentioned that the idea of assumption 4.1 is that the state-action pair will be visited infinitely often. However, in the definition of MDP, it is said that there is a sink state, and the MDP is transient for any policy. Do these two statements conflict with each other? Please correct me if I’m wrong.

Moreover, the discussion on the related work on risk-averse total reward MDP is missing. The authors list a series of works on the risk-averse RL or total reward MDP, but there is no comparison between this paper and the literature.

Clarity: 2

The entire writing of this paper is OK. But it is confusing in Algorithm 1 why the elicitability property can be replaced with a single step of gradient descent. Following the authors’ writing, I thought we should solve Eq. 11 in each iteration. However, only a single step is adopted in each iteration. The authors answer the question in Lemma B.8 and Eq. 32 in the appendix. I strongly recommend that the authors move these parts to the main text so that it would be clear that a single-step gradient descent is equivalent to one step of Q-learning.

Significant: 3

The risk-averse RL is an important topic in Reinforcement Learning. The authors put forward a tabular Q-learning model-free algorithm to solve the risk-averse RL in total reward MDP, which could be an important step for future work on more complex environments.

Originality:2

The technical novelty of this paper is somewhat limited. Firstly, the ERM Bellman operator and Bellman optimal equation are from the previous work [41, theorem 3.3]. In order to solve the unknown transition, the authors utilized the elicitability property. However, according to Lemma B.8 and Eq. 32 in the appendix, the proposed algorithm can be considered as a Q-learning algorithm, and thus, the convergence is guaranteed by the standard convergence analysis for stochastic iterative algorithms.

---

> ### Author Rebuttal · Authors · 2025-07-30
>
> Rebuttal:
>
> We sincerely appreciate the detailed review and perspective.
>
> ***I have a question about the assumption. In the paper, the authors mentioned that the idea of assumption 4.1 is that the state-action pair will be visited infinitely often. However, in the definition of MDP, it is said that there is a sink state, and the MDP is transient for any policy. Do these two statements conflict with each other? Please correct me if I’m wrong.***
>
> Thanks for raising this point. These statements do not conflict with each other, and we will clarify this point in the paper. For Q-learning to converge, it requires data from multiple episodes. Even though an individual episode terminates upon reaching the sink state, the agent must restart and revisit state-action pairs across multiple episodes, satisfying the infinite visitation requirement for convergence.
>
>
> ***The discussion on the related work on risk-averse total reward MDP is missing. The authors list a series of works on the risk-averse RL or total reward MDP, but there is no comparison between this paper and the literature.***
>
> Thank you. To the best of our knowledge, this is the first paper that studies Q-learning for the risk-averse total reward criterion. Prior risk-averse RL work focuses either on model-based dynamic programs [41] or the average reward, finite-horizon, or discounted criterion [23, 27, 17]. We will emphasize this distinction in the final version of the paper.
>
>
> ***But it is confusing in Algorithm 1 why the elicitability property can be replaced with a single step of gradient descent. Following the authors’ writing, I thought we should solve Eq. 11 in each iteration. However, only a single step is adopted in each iteration. The authors answer the question in Lemma B.8 and Eq. 32 in the appendix. I strongly recommend that the authors move these parts to the main text so that it would be clear that a single-step gradient descent is equivalent to one step of Q-learning.***
>
> Thanks! We will move Lemma B.8 and Eq.32 to the main text in the final version. We will also add a detailed sketch of the proof of Lemma B.9 to the main text, clarifying the relationship between the elicitability property, the monotonicity property, single-step gradient descent, and one step of Q-learning in the final version.
>
> ***The technical novelty of this paper is somewhat limited. Firstly, the ERM Bellman operator and Bellman optimal equation are from the previous work [41, theorem 3.3]. To solve the unknown transition, the authors utilized the elicitability property. According to Lemma B.8 and Eq. 32 in the appendix, the proposed algorithm can be considered as a Q-learning algorithm. Thus, the convergence is guaranteed by the standard convergence analysis for stochastic iterative algorithms.***
>
> We take this concern seriously but believe that our algorithm and its technical analysis depart significantly from standard Q-learning approaches. To develop the algorithm, we leverage the elicitability property of ERM. That is, we need two definitions of the ERM Bellman operator and Bellman optimal equation. First, for each state-action pair, we define the standard ERM Bellman operator in Eq. (7) and the Bellman optimal equation in Eq. (8). Their correctness can be verified from [41, theorem 3.3]. Second, we leverage the elicitability property of ERM to define a new Bellman operator in Eq. (11) as the minimizer of the exponential loss function. We then prove the equivalence of the two ERM Bellman optimal equations, as stated in Theorem 3.3 of our paper. This demonstrates the correctness of our new ERM Bellman operator and Bellman optimal equation, which are essential to developing the proposed ERM-TRC and EVaR-TRC Q-learning algorithms.
>
> The main challenge in the technical analysis is that under the total reward criterion, the ERM Bellman operator is not a contraction captured by (see Assumption B.2 in the paper). The lack of the contraction property prevents the direct application of the results from the standard Q-learning and the risk-averse total reward setting, which our paper addresses. Instead, we need to use other properties of the Bellman operator to prove the convergence of the proposed Q-learning algorithms. In addition, under the total reward criterion, q-value functions can be unbounded. Then, an additional bounded condition on temporal difference(TD) is needed to bound the q-value. This work uses the monotonicity of the Bellman operator and a separate bounded condition to prove the convergence of the proposed Q-learning algorithms.

---

> > ### Comment · Reviewer_27u3 · 2025-08-05
> >
> > Thanks for the detailed response. I think the authors have answered my first three questions well and given a detailed explanation of novelty. After consideration, I think the equivalence of the elicitability of ERM is somehow novel and important to the proposed algorithm. Thus, I would like to increase my rating.

---

### Decision · Program_Chairs · 2025-09-17

**Decision:**

Accept (poster)

**Comment:**

This paper studies risk averse RL for ERM and EVAR risk metrics under the total reward formulation. A Q-learning-like model-free algorithm is proposed and extensive theoretical analysis is provided. Small scale empirical results are also provided to support the usefulness of the proposed algorithm. Overall, the reviewers agree that this paper studies an important problem and the approach is technically solid. I, therefore, recommend accept.